# The LRR-TM protein PAN-1 interacts with MYRF to promote its nuclear translocation in synaptic remodeling

**Shi-Li Xia[1,2†], Meng Li[1,2†], Bing Chen[2], Chao Wang[2], Yong-Hong Yan[3], Meng-Qiu Dong[3], Yingchuan B Qi[1,2]\***

[1]School of Life Science and Technology, ShanghaiTech University, Shanghai, China; [2]College of Life and Environmental Sciences, Hangzhou Normal University, Hangzhou, China; [3]National Institute of Biological Sciences, Beijing, China

**Abstract** Neural circuits develop through a plastic phase orchestrated by genetic programs and environmental signals. We have identified a leucine-rich-repeat domain transmembrane protein PAN-1 as a factor required for synaptic rewiring in *C. elegans*. PAN-1 localizes on cell membrane and binds with MYRF, a membrane-bound transcription factor indispensable for promoting synaptic rewiring. Full-length MYRF was known to undergo self-cleavage on ER membrane and release its transcriptional N-terminal fragment in cultured cells. We surprisingly find that MYRF trafficking to cell membrane before cleavage is pivotal for *C. elegans* development and the timing of N-MYRF release coincides with the onset of synaptic rewiring. On cell membrane PAN-1 and MYRF interact with each other via their extracellular regions. Loss of PAN-1 abolishes MYRF cell membrane localization, consequently blocking *myrf*-dependent neuronal rewiring process. Thus, through interactions with a cooperating factor on the cell membrane, MYRF may link cell surface activities to transcriptional cascades required for development.

**\*For correspondence:**
qiyc@shanghaitech.edu.cn

[†]These authors contributed equally to this work

**Competing interests:** The authors declare that no competing interests exist.

## Introduction

During development, neurons enter a plastic phase, during which synapses can be added or pruned until mature circuits are formed (*Purves and Lichtman, 1980*). It is known that the plastic process is coordinated by genetic programs and environmental signals (*Espinosa and Stryker, 2012*), but the underlying mechanisms are not fully understood. A striking and informative example of circuit plasticity occurs during the development of the *C. elegans* nervous system, in which six GABAergic motor neurons, known as dorsal D (DD) neurons, reverse their axonal and dendritic domains at the end of larval stage 1 (L1), with no overt changes in morphology (*Hallam and Jin, 1998*; *White et al., 1978*). After completion of the synaptic rewiring (or remodeling), DD neurons acquire new connectivity within the motor circuit. This unique event provides an attractive system for identifying the molecular requirements of circuit remodeling (*Jin and Qi, 2018*; *Kurup and Jin, 2016*).

Synaptic rewiring of DD neurons is driven by *myrf*, *Caenorhabditis elegans* ortholog of mammalian *Myrf* (*My*elin *r*egulatory *f*actor) (*Meng et al., 2017*). *Myrf* family consists of conserved, membrane-associated transcription factors. Evidence from studies in model organisms (including humans) collectively indicates an essential role for *Myrf* in animal development. In mice, germline deletion of *Myrf* led to early embryonic lethality (*Bujalka et al., 2013*), and oligodendrocyte-specific deletion of *Myrf* blocked myelin formation (*Emery et al., 2009*; *Koenning et al., 2012*). Deletion of *mrfA*, the ortholog of *Myrf* in *Dictyostelium*, caused delayed multicellular development and deficient pre-stalk cell differentiation (*Senoo et al., 2012*). The null mutation of *C. elegans myrf-1* resulted in developmental arrest at the end of L1, and a partial loss of function mutant of *myrf-1* displayed an extra L4-adult molt, indicating that *myrf-1* is involved in multi-stage development (*Meng et al., 2017*;

*Russel et al., 2011*). Over several years, more than 30 human genetic variants of *MYRF* have been identified in patients as causes of developmental anomalies in a range of tissues, including heart, lungs, diaphragm, genitourinary tract, and eyes (*Qi et al., 2018*; *Rossetti et al., 2019*). In these cases, human *MYRF* appeared to be haploinsufficient. Despite its essential roles, how *Myrf* regulates animal development is largely uncharacterized.

The founding member of the *Myrf* family, mouse *Myrf*, transcriptionally regulates expression of myelinogenesis-related genes in oligodendrocytes (*Bujalka et al., 2013*; *Li et al., 2013*). The transcriptional activity is conferred by an N-terminal fragment of MYRF (N-MYRF), which harbors an *I*mmunoglobulin (Ig)-fold like DNA-binding domain (DBD) homologous to the DBD of Ndt80, a pleiotropic transcription factor in yeast (*Bujalka et al., 2013*; *Chen et al., 2018*; *Li et al., 2013*; *Zhen et al., 2017*). At least in cultured cells, full-length MYRF is localized on ER membrane by its single-pass transmembrane domain and undergoes self-catalyzed proteolysis to release N-MYRF (*Bujalka et al., 2013*; *Kim et al., 2017*; *Li et al., 2013*). Cleavage is mediated by an intramolecular chaperone of endosialidase (ICE) domain, located carboxyl to the cleavage point (*Schwarzer et al., 2007*). The ICE domain was initially found in a tailspike protein (endosialidase) of bacteriophages. Assembly of an ICE domain homotrimer catalyzes cleavage, releasing the endosialidase activity region, which in turn breaks down bacterial cell wall (*Schulz et al., 2010*). MYRF is the only eukaryotic protein known to contain the ICE domain, and in MYRF the domain appears to have adopted its ancestral role—to mediate MYRF trimerization and catalyze cleavage (*Kim et al., 2017*; *Li et al., 2013*; *Meng et al., 2017*; *Senoo et al., 2013*). Cleavage of MYRF on the ER-membrane appears to be constitutive in cultured cells, for which the MYRF ICE domain has been renamed as an intramolecular chaperone auto-cleavage (ICA) domain (*Li et al., 2013*).

Liberating active form of transcription factors from membrane is of exquisite cellular importance, as the process provides a direct portal for cells to respond to internal or environmental cues by changing gene expression programs. Examples are the processing events that govern sterol regulation, Notch signaling, and the unfolded protein response (*Hoppe et al., 2001*; *Seo et al., 2008*). Transmembrane motifs are well conserved among MYRF family proteins, implying that MYRF's membrane association confers important function; yet the cleavage of mammalian MYRF appears to occur constitutively, which is unusual because for all other known membrane-bound transcription factors, their release from membranes are regulated by inductive signals. It thus remains unclear why MYRF must go onto the ER membrane, and if the release of N-MYRF from ER is regulated *in vivo*.

Synaptic rewiring in *C. elegans* is also dependent on the cleavage and nuclear function of MYRF (*Meng et al., 2017*), suggesting that the molecular activity of MYRF is conserved. Curiously, while over-expression of N-MYRF-1 drives precocious synaptic rewiring, over-expression of full-length MYRF-1 does not. When MYRF was over-expressed in *C. elegans*, even by modest quantities, MYRF protein accumulated in a cytoplasmic structure (likely ER) without increasing in the nucleus (*Meng et al., 2017*; *Russel et al., 2011*), suggesting that MYRF nuclear localization does not automatically occur. In addition, when expressed in human cell lines, *Ce* MYRF is localized to the ER but cannot be cleaved (*Meng et al., 2017*). These observations together suggest that cleavage of *Ce* MYRF is not constitutive, but instead controlled by putative additional factors or dependent on specific conditions.

Because MYRF-1 and its paralog MYRF-2 are together indispensable for driving the synaptic rewiring (*Meng et al., 2017*), unraveling the regulation of MYRF activity is key to understanding this plastic process. We herein identify a leucine-rich repeat transmembrane (LRR-TM) protein PAN-1 as an essential factor in promoting synaptic remodeling. PAN-1 is a type I transmembrane protein with extracellular leucine-rich repeat (LRR) domains. PAN-1 was previously found to be a component of P-granule (an RNA granule in germline), expressed broadly in somatic tissues, and required for larval development (*Gao et al., 2012*; *Gissendanner and Kelley, 2013*). We find that PAN-1 localizes on cell membrane and interacts with MYRF. PAN-1 is required for MYRF's cell membrane localization and nuclear translocation. Our findings reveal undescribed activities of MYRF on cell membrane, which enact critical developmental regulation of synaptic rewiring.

## Results

### The LRR-TM protein PAN-1 is a binding factor of MYRF

To identify factors that potentially interact with MYRF and regulate MYRF's activity, we performed mass-spectrometry analysis of co-immunoprecipitation (IP-MS) of GFP::MYRF-1 as well as GFP::MYRF-2 from extracts of transgenic animals expressing *GFP::myrf* driven by endogenous *myrf* promoters (*Meng et al., 2017*). We identified an LRR-TM protein PAN-1 (M88.6) as a factor with high binding specificity to both MYRF-1 and MYRF-2 (*Figure 1A*). Inversely, when PAN-1$^{GFP}$ (a *pan-1$^{GFP}$* knock-in allele) was pulled down using similar procedures, MYRF-1 and MYRF-2 were found among the most specific interactors of PAN-1. The results of the three IP-MS analyses indicate that MYRF and PAN-1 can form a protein complex *in vivo*.

We next tested if MYRF and PAN-1 interact *in vitro*. When both proteins were expressed in HEK293T cells, MYRF-1 could be co-immunoprecipitated with PAN-1 (*Figure 1B*). Because *C. elegans* proteins were expressed in human cells in this experiment, it is likely that MYRF and PAN-1 interact directly. To be indirectly bound, there would need to be an additional human protein that could interact with both proteins to form the putative complex, which seems unlikely considering the evolutionary distance between humans and nematodes.

### Loss of *pan-1* blocks synaptic remodeling

Because MYRF-1 and MYRF-2 have been identified as critical, positive regulators of synaptic remodeling, we examined if loss of *pan-1* affects synaptic remodeling. We analyzed the synaptic remodeling in *pan-1(gk142)*, a null deletion allele (*Gao et al., 2012*). Synaptic remodeling of DD neurons normally occurs at late L1 when new synapses form in the dorsal processes of DD neurons, which is marked by the dorsal appearance of clusters of synaptic vesicles (*Figure 1C–E*; *Hallam and Jin, 1998*). By late L1, few synapses had formed in the dorsal processes of DDs in *pan-1(gk142)* mutants (*Figure 1D,E*), indicating that the synaptic remodeling was defective in *pan-1* mutants. Axon morphology of DDs in *pan-1(gk142)* was no different from control animals; if anything, the GFP intensity of dorsal processes in *pan-1(gk142)* appeared to be weaker (*Figure 1F*). The defective synpatic rewiring had not improved by the end of L2 (28 post-hatch hours) (*Figure 1D*). The ventral synapses of DDs in *pan-1(gk142)*, labeled by synpatic vesicle clusters, appeared to be similar in puncta brightness and number, compared to wild-type animals. The presence of juvenile DD synapses in *pan-1* mutants suggests that the loss of *pan-1* does not generally affect synaptogenesis in DDs. Because *pan-1* mutants are the first generation from heterozygous parent, it remains possible that maternal PAN-1 may contribute to embryonic and early L1 development of *pan-1* mutants. Nevertheless, the absence of zygotic *pan-1* results in synaptic remodeling defects that are similar to those when *myrf* is lost (*Meng et al., 2017*).

### *pan-1* regulates DD synaptic remodeling cell-autonomously

*pan-1* is expressed broadly in embryos and larvae (*Gao et al., 2012*; *Gissendanner and Kelley, 2013*). To confirm that loss of *pan-1* is responsible for synaptic remodeling defects in *pan-1* mutants, we performed rescue experiments by expressing *pan-1* in DD neurons (*Figure 2A,B*; *Figure 2—figure supplement 1A,B*). Among three isoforms of *pan-1*, the longest transcript, *pan-1b*, encodes a 594 amino acid (aa) protein containing 14 leucine rich repeat (LRR) domains, a transmembrane domain, and an N-terminal ER-targeting signal peptide (SP) (*Figure 2A*; *Gao et al., 2012*). The *pan-1a* and *pan-1c* isoforms encode shorter peptides that lack residues 1–309 and 1–502 of the *pan-1b* coding sequence, respectively. PAN-1B is predicted to be a type I transmembrane protein. In contrast, PAN-1A and PAN-1C are unlikely to be inserted into membrane because they lack the signal peptide. Expression of longest isoform *pan-1b* driven by *unc-25$_{pro}$* (expressed in DD and VD) restored the synaptic remodeling defects in *pan-1* mutants, while the other two isoforms, *pan-1a* or *pan-1c* could not (*Figure 2B*). Furthermore, expression of *pan-1* (referring to *pan-1b* hereafter) driven by *flp-13$_{pro}$* (expressed in DD not VD among ventral cord motor neurons) also rescued synaptic rewiring defects in *pan-1* mutants (*Figure 2—figure supplement 1A*). In contrast, specific expression of *pan-1* in muscles, epidermis, or intestine did not mitigate the deficient synaptic remodeling in *pan-1(0)* (*Figure 2—figure supplement 1B*). Together, these data demonstrate that a transmembrane isoform of PAN-1, PAN-1B, regulates synaptic remodeling in DDs cell autonomously; in other

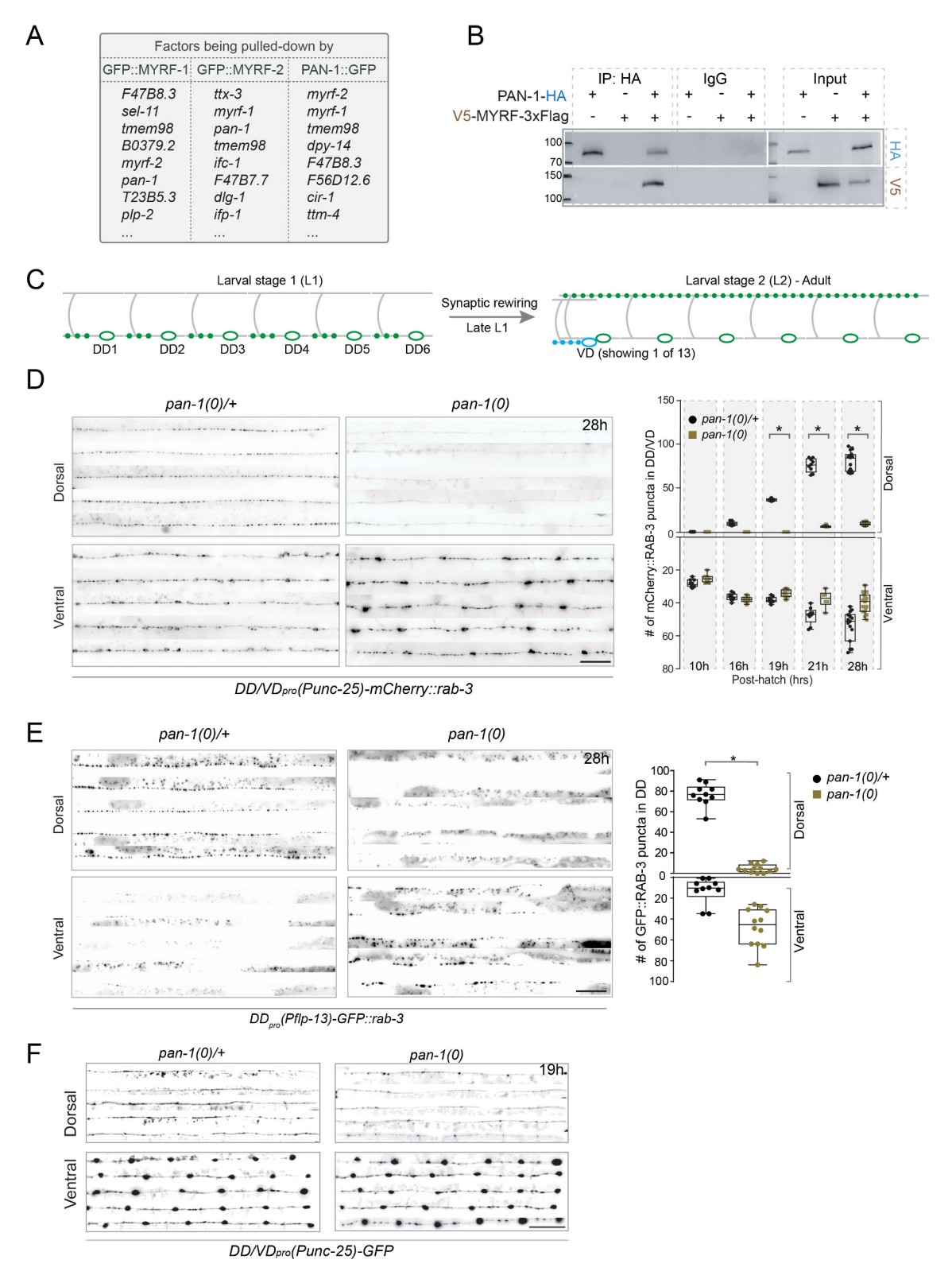

**Figure 1.** *pan-1* mutants exhibit blocked synaptic rewiring. (**A**) Top of list for factors interacting with MYRF-1, MYRF-2, and PAN-1, identified by co-IP and Mass-Spec analysis. The factors are ranked by specificity of the binding. The strains used are *myrf-1(ju1121); myrf-1_{pro}-GFP::myrf-1::Flag (ybqEx164)*, *myrf-2(ybq42); myrf-2_{pro}-GFP::myrf-2 (ybqIs128)*, and *pan-1^{GFP} (ybq47)*. Complete list of IP-MS results can be found in **Supplementary file 1** for MYRF-1, **Supplementary file 2** for MYRF-2, and **Supplementary file 3** for PAN-1. (**B**) Co-immunoprecipitation of PAN-1 and MYRF in HEK293T cells. Rabbit anti-

*Figure 1 continued on next page*

*Figure 1 continued*

HA was used in co-IP, and the target bands in Western Blot were visualized using murine anti-HA and murine anti-V5. (**C**) Illustration of synaptic rewiring in six DDs. The presynaptic sties, indicated by green dots, are localized in ventral processes at L1, and become localized in dorsal processes after the rewiring is complete. The rewiring begins from late L1 and is complete by late L2. Ventral D (VD) is a class of GABAergic neurons analogous to DDs and born at late L1. Presynaptic sites of VDs form in their ventral processes. (**D**) Deficient synaptic remodeling in *pan-1(gk142)* mutants. The presynaptic sites in DD/VD are labeled by *unc-25*$_{pro}$*-mCherry::rab-3(juIs236)*. Images of five ventral and dorsal cords are vertically tiled. At 28 hr, the synaptic remodeling was complete in control animals *pan-1(gk142)/mT1*, marked by the formation of dorsal synapses. In contrast, there was few clear clusters in dorsal cord of *pan-1* mutants. Scale bar, 20 μm. In quantification graph, the number of synapses in DD/VD of *pan-1(gk142)* and control animals *pan-1 (gk142)/mT1* were counted, which is shown as Mean ± SEM (t test. *, p<0.001). (**E**) Deficient synaptic remodeling in *pan-1(gk142)* mutants. The presynaptic sites in DD are labeled by *flp-13*$_{pro}$*-GFP::rab-3(ybqIs46)*. Images of five ventral and dorsal cords are vertically tiled. At 28 hr, the synaptic remodeling was complete in control animals *pan-1(gk142)/mT1*, marked by the formation of dorsal synapses and disappearance of ventral synapses. In contrast, there was few clear clusters in dorsal cord of *pan-1* mutants while ventral synapses were retained. Scale bar, 20 μm. In quantification graph, the number of synapses in DD of *pan-1(gk142)* and control animals *pan-1(gk142)/mT1* were counted, which is shown as Mean ± SEM (t test. *, p<0.001). (**F**) Axon morphology of DD neurons is normal in *pan-1(gk142)* mutants. The axons were labeled by *unc-25*$_{pro}$*-GFP(juIs76)*. It is normal that a gap is present between dorsal processes of DD3 and DD4 in a subset of animals of this stage. More than 15 animals were examined for each genotype. Images of five ventral and dorsal cord were vertically tiled. Scale bar, 20 μm.

words, restoring PAN-1 in DDs is sufficient to promote synaptic rewiring in developmentally arrested mutants.

*pan-1* mutants exhibit progressive larval arrest and never grow to L4 or fertile adults. Arrested animals feed, move, and live a normal life span that is similar to wild type (*Gao et al., 2012*). To address if blocked synaptic remodeling in *pan-1(0)* is caused by general developmental arrest, we generated a cell-type-specific deletion of *pan-1* by combining the *pan-1*$^{LoxP}$ allele and a DD-expressing *Cre* line (*Figure 2A,C*; *Figure 2—figure supplement 1C*). The animals with the compound allele grew and reproduced like wild type, such that we could analyze possible synaptic remodeling defects in otherwise wild-type animals. We observed the disappearance of synaptic markers in segments of dorsal cord in many *Cre*-expressing animals, while similar gaps were not observed in non-*Cre* animals. The axon morphology of DDs in these animals was not altered (*Figure 2—figure supplement 1C*). Because the span of the missing segments closely corresponded with the position of dorsal neurites of individual DD neurons (*Figure 1C*), the lack of dorsal clusters was likely due to failed synaptic remodeling in corresponding DD neurons. For those DD neurons that displayed failed synaptic remodeling, the deficiency persisted into L4 (*Figure 2—figure supplement 1C*) and adult animals. The incomplete penetrance of rewiring defects in these animals may be attributed to the short time period from the expression of *Cre* to the initiation of DD remodeling (~15 hr), such that some DD neurons may have acquired sufficient amount of PAN-1 for synaptic remodeling. Nevertheless, the data strongly suggest that the neuronal remodeling deficiency in *pan-1(0)* is neither a secondary result of overall developmental arrest in mutants, nor due to developmental delay. The results also further support a cell autonomous function of *pan-1* in DDs.

## PAN-1 localizes on cell membrane

We investigated where PAN-1 is subcellularly localized. In *pan-1*$^{GFP}$ knock-in allele the GFP tag was inserted at the C-terminal of the *pan-1* coding sequence. PAN-1$^{GFP}$ signal was detected broadly in embryos and early larvae, but decreased substantially in late larvae and adults (*Figure 3A*; *Figure 3—figure supplement 1*). Two types of PAN-1$^{GFP}$ subcellular patterns were observed: at the cell membrane and as cytoplasmic puncta. Some prominent PAN-1$^{GFP}$ signal was found in seam cells, a type of epidermal cells that shows stem-cell like division during larval development. These cells were lined up in a single profile at each lateral line and embedded in syncytial epidermis. Between the seam cells and the epidermis, a ring of tight junctions formed on the apical-lateral side of the cells. By colabelling a tight-junction component DLG-1::RFP and PAN-1$^{GFP}$ (*Figure 3B*), we observed that PAN-1 distributed at the lateral basal membrane of seam cells. PAN-1 is also localized on the cell membrane of many ventral cord neurons, including DD neurons at L1 and L2 (*Figure 3C, D*). Therefore, we observed that PAN-1 is localized on the cell membrane.

To examine if the membrane localization of PAN-1 is critical for function, we expressed deletion variants of PAN-1 protein in DDs and looked for restoration of blocked synaptic remodeling in *pan-1 (0)* (*Figure 3E*; *Figure 3—figure supplement 2*). Deleting either the SP or TMD caused PAN-1 to lose its rescuing activity, suggesting that PAN-1's membrane localization is required for its function.

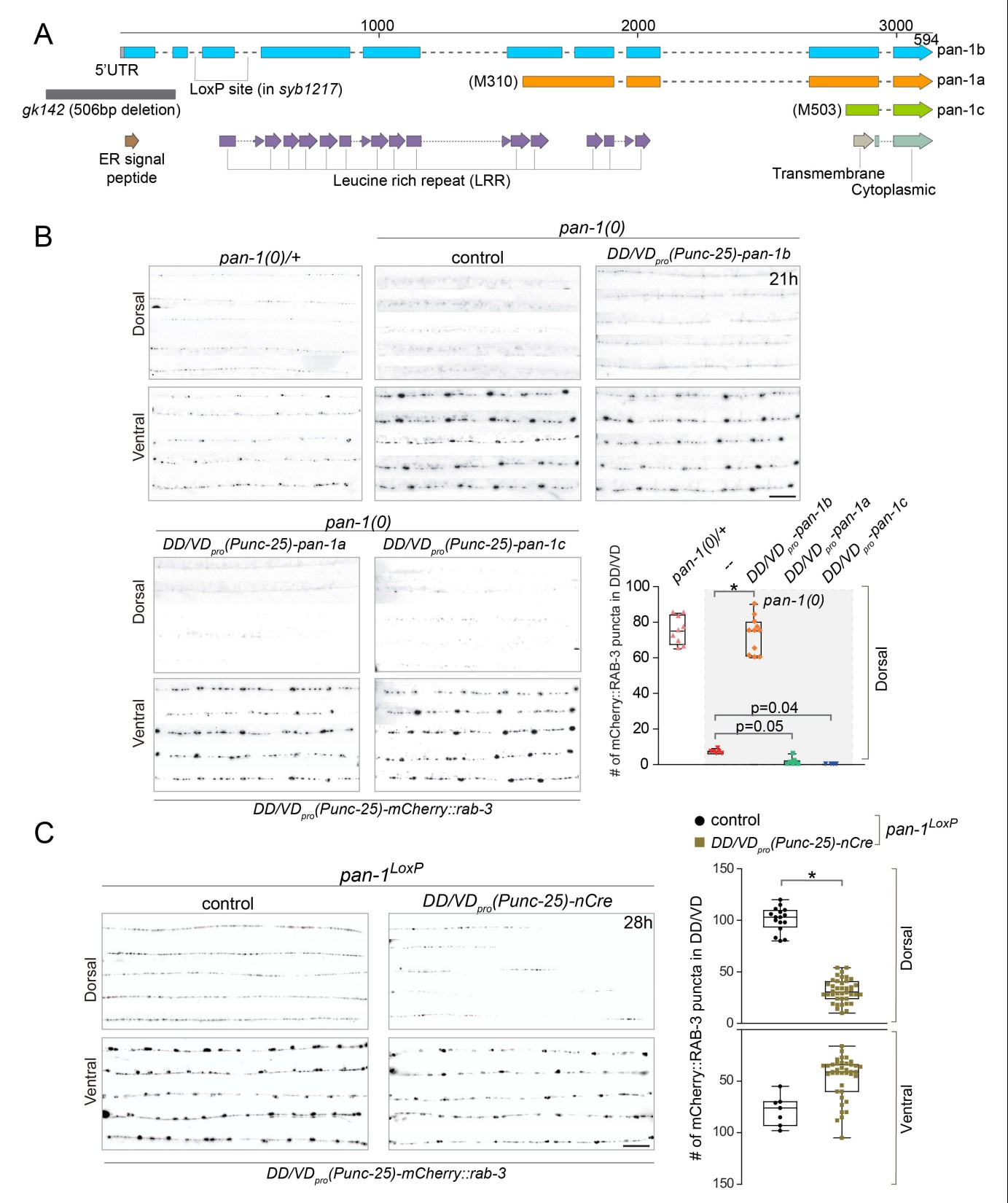

**Figure 2.** PAN-1 functions cell-autonomously in synaptic rewiring. (**A**) Illustration of *pan-1* gene and protein domains. Three *pan-1* transcripts are produced *via* trans-splicing. *gk142* deletion eliminates the promoter and all three transcripts. The insertion sites of *LoxP* insertion in *pan-1(syb1217)* are

*Figure 2 continued*

indicated. (**B**) Deficient synaptic remodeling in *pan-1(gk142)* was rescued by DD/VD-expressing *pan-1b*, but not *pan-1a* or *pan-1c*. The presynaptic sites in DD/VD are labeled by *unc-25_{pro}-mCherry::rab-3(juIs236)*. The stage of animals was 21 post-hatch hours. Images of five ventral and dorsal cords are vertically tiled. Scale bar, 20 μm. The number of synapses in DD/VD of genotype-indicated animals was counted and shown as Mean ± SEM (t test. *, p<0.001). Expressed were three isoforms of *pan-1*: *unc-25_{pro}-pan-1b::GFP(ybqIs131); unc-25_{pro}-pan-1a(ybqEx624); unc-25_{pro}-pan-1c(ybqEx626)*. (**C**) DD/VD-specific gene inactivation of *pan-1*. *pan-1^{LoxP}(syb1217)* animals with or without DD/VD-expressed *nCre* transgene, *unc-25_{pro}-nCre(tmIs1073)*. Two *LoxP* sequences are inserted inside the flanking introns of the third exon of *pan-1* gene. Animals are labeled by presynaptic marker in DD/VD, *unc-25_{pro}-mCherry::rab-3(juIs236)*. In the presence of *nCre*, segments of synapses are missing in dorsal cords. Images of five ventral and dorsal cords are vertically tiled. Scale bar, 20 μm. The number of synapses in DD/VD of genotype-indicated animals was counted and shown as Mean ± SEM (t test. *, p<0.001).

The online version of this article includes the following figure supplement(s) for figure 2:

**Figure supplement 1.** PAN-1 acts cell-autonomously.

We found that the cytoplasmic region of PAN-1 was not essential for its function; in contrast, the LRR domains, the main motifs within the extracellular region, were necessary for PAN-1's activity in synaptic remodeling. Together, these data imply that PAN-1's cell-membrane localization and its extracellular domain are important.

## MYRF is also trafficked to cell membrane

The full-length MYRF was previously characterized to be localized on ER in cultured mammalian cells. The interaction between PAN-1 and MYRF raised the question where MYRF is subcellularly localized in developing *C. elegans*. In *myrf-1^{GFP}* knock-in allele, the GFP was inserted into the N-terminal region of MYRF, such that GFP labels both the full-length and N-terminal MYRF (**Figure 4A**). The fusion protein was fully functional as knock-in animals grew and behaved like wild type. The native GFP signal was weak but could be detected using sensitive imaging tools, such as sCMOS camera and Zeiss Airyscan. In early L1 larvae, GFP signal was observed at the cell membrane of many cell types, including pharynx, epidermis, and neurons, but were not obviously enriched in the cytoplasm or nucleus (**Figure 4B,C**). Toward late L1 stage, GFP signals started to appear in the nucleus. Within a stage-synchronized population, animals exhibited variable patterns of MYRF-1^{GFP}'s localization, that is primarily at the cell membrane, primarily in the nucleus, or both at the cell membrane and in the nucleus (**Figure 4B–F**). By colabeling with DD-expressing mCherry::Rab-3, we observed cell-membrane-to-nucleus transition of MYRF-1^{GFP} in DD neurons (**Figure 4E,F**). Quantification of these patterns revealed a trend of progressively increasing MYRF-1^{GFP} nuclear localization (**Figure 4C**). A similar early-stage cell membrane-localization was observed for MYRF-2^{GFP}, the paralog of MYRF-1, except that MYRF-2^{GFP} exhibited stronger signals than MYRF-1^{GFP} in early L1 larvae (**Figure 4—figure supplement 1**). MYRF-2^{GFP} also displayed the trend of localization transitioning from cell membrane to nucleus during L1 development. Together, we observed that MYRF^{GFP} is localized primarily at cell membrane in early L1 animals, and becomes increasingly localized in the nucleus as larval development progresses.

To test if the cell membrane is where N-MYRF is cleaved and released, we generated a loss-of-cleavage mutant allele of *myrf-1* by changing two critical catalytic residues within the ICE domain, which catalyzes the cleavage of MYRF (**Figure 5A**; **Kim et al., 2017**; **Li et al., 2013**). We expected that full-length mutant MYRF-1 would accumulate as it could not be cleaved. This *myrf-1^{S483A K488A GFP}* allele caused a larval arrest in animals (**Figure 5B,C**), similar to the arrest observed in *myrf-1* null mutants and consistent with our previous finding that *myrf-1*'s activity is cleavage-dependent. Importantly, MYRF-1^{S483A K488A GFP} was only detected at the cell membrane but not in the nucleus at any viable stages (**Figure 5D–G**). The intensity of MYRF-1^{S483A K488A GFP} was increased compared to wild type, in accordance with our prediction that mutant MYRF would accumulate on cell membrane because it cannot be cleaved. The overall course of synaptic rewiring in *myrf-1^{S483A K488A GFP}* was similar to *myrf-1(ybq6)*, a putative null indel allele, although the synaptic rewiring in *myrf-1^{S483A K488A GFP}* appeared to be less complete than *myrf-1(ybq6)* (**Figure 6**; **Figure 6—figure supplements 1–6**). When *myrf-2*, the paralog of *myrf-1*, was simultaneously mutated, the synaptic rewiring was severely blocked (**Figure 6**; **Figure 6—figure supplements 7–12**), consistent with our previous finding that *myrf-1* and *myrf-2* are together indispensable for synaptic rewiring in DDs. These data

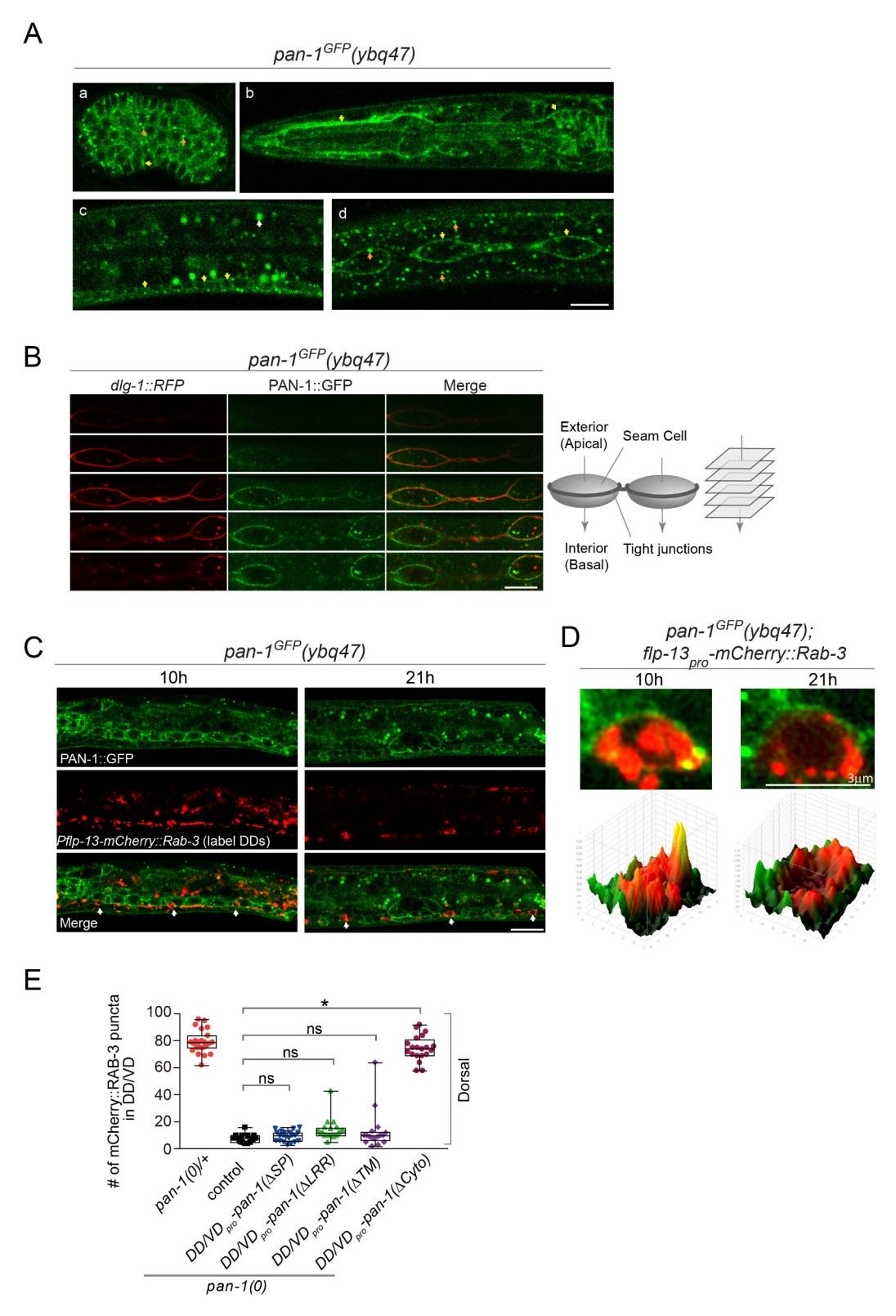

**Figure 3.** PAN-1 localizes on the cell membrane. (**A**) Signals of PAN-1<sup>GFP</sup> in *pan-1<sup>GFP</sup>(ybq47)* (knock-in) animals. a, embryo. b, head region of L2 larva. c and d, trunk region of L2 larva with a focus on ventral cord neurons (yellow arrows) and seam cells (yellow arrows), respectively. Yellow arrow, GFP at the cell membrane. Orange arrow, GFP as cytoplasmic puncta. White arrow, auto-fluorescence. Images are single optical slices by confocal microscopy. Scale bar, 10 μm. (**B**) Localization of PAN-1<sup>GFP</sup>(*ybq47*) in respect to tight junction protein DLG-1::RFP(*mcIs46*) in seam cells. Shown are a consecutive

*Figure 3 continued on next page*

*Figure 3 continued*

series of confocal optical sections (0.36-μm-thick section at 0.7 μm interval on Z-axis). PAN-1 partially overlaps with DLG-1, but primarily localizes at basal lateral membrane of seam cell (towards the interior). Scale bar, 10 μm. (C) Colocalization of PAN-1$^{GFP}$ and DD neuron marker. The animals *pan-1$^{GFP}$(ybq47); Pflp-13-mCherry::rab-3(ybqIs1)* at 10, 21 post-hatch hours were imaged by Airyscan confocal microscopy. The animals carry *glo-1(zu391)* to reduce auto-fluorescence. RAB-3 is localized in cytoplasm, while PAN-1 is enriched on the cell membrane. White arrow, soma of DDs. Scale bar, 10 μm. (D) Shown are the magnified images of DD soma from the colabeling experiment (C), and 3-D intensity plots of green and red signals in the upper image. (E) Quantification of the synaptic remodeling in *pan-1(gk142)* carrying transgenes of DD/VD-expressed *pan-1* variants. The number of synapses in DD/VD of animals of indicated genotypes were counted and shown as Mean ± SEM (t test. ns, p>0.05. *, p<0.001.). Expressed in the transgenes are *unc-25$_{pro}$-pan-1ΔSP(ybqEx643); unc-25$_{pro}$-pan-1ΔLRR(ybqEx642); unc-25$_{pro}$-pan-1ΔTM (ybqEx644); unc-25$_{pro}$-pan-1ΔCyto(ybqEx645)*.

The online version of this article includes the following figure supplement(s) for figure 3:

**Figure supplement 1.** *pan-1* expression is temporally regulated during development.
**Figure supplement 2.** LRR region and TMD of PAN-1 are important for synaptic rewiring.

support that MYRF is associated with cell membrane before cleavage, which is temporally controlled during development.

## PAN-1 and MYRF interact through extracellular domains

We next characterized how MYRF and PAN-1 interact with each other using human cell lines (*Figure 7*). PAN-1 is a Type-I transmembrane protein with an ER-targeting SP. The region that contains the LRR domains, is predicted to be non-cytoplasmic, that is in the ER-lumen or extracellular. There is also a short cytoplasmic region in PAN-1. MYRF is inserted in membrane by its single transmembrane domain, and 1–482 region of MYRF (responsible for transcriptional activation) is cytoplasmic, while 680–931 region of MYRF (of unknown function) is lumenal or extracellular. When PAN-1 and MYRF were co-expressed in HEK293T cells, both proteins were primarily detected in the ER (data not shown). PAN-1 binding was observed with the C-terminal region of MYRF, but not with the N-terminal region (*Figure 7B*). This result is consistent with their topographic configuration on ER membrane, which predicts that PAN-1 may use its non-cytoplasmic LRR domains to interact with the non-cytoplasmic region of MYRF.

To further test which part of PAN-1 is required for MYRF interaction, we generated a series of truncated PAN-1 mutants and analyzed their ability to bind MYRF (*Figure 7A,C*). The LRR domains of PAN-1 was responsible for binding with C-MYRF (*Figure 7C*); the binding occurred when the two fragments coexisted either in the ER lumen or cytoplasm of HEK cells. The cytoplasmic region of PAN-1 was dispensable for PAN-1's binding with MYRF. When the SP was deleted from PAN-1, the protein could not be detected by western blot, possibly due to being insoluble as it carries a hydrophobic TMD. Together these data indicate that PAN-1 uses its LRR domain to interact with the extracellular region of MYRF.

## Extracellular domains of MYRF are required for its normal function

Because PAN-1 and MYRF interact via their extracellular domains, we tested if the extracellular region of MYRF is necessary for its cell membrane localization and function. We generated an in-frame deletion allele of *myrf-1$^{1-700\ GFP}$* that removed the entire extracellular domain from MYRF-1 (*Figure 8A*). This deletion allele was recessive and homozygous mutants exhibited larval arrest at L2, indicating that the extracellular domain of MYRF-1 was necessary for normal function (*Figure 8B,C*). Importantly, MYRF-1$^{1-700\ GFP}$ displayed prominent cytoplasmic signals while its cell-membrane or nuclear localization were not obvious at any viable stages (*Figure 8D–F*; *Figure 8—figure supplement 1*), showing that the extracellular domain is required for MYRF's proper localization.

One segment within the extracellular domain of MYRF is designated as the C2 domain (conserved domain 2). We tested if the C2 domain is particularly important for MYRF function (*Figure 8A*) by generating another in-frame deletion allele of *myrf-1$^{1-790\ GFP}$* that removed the C2 domain (791-926). *myrf-1$^{1-790\ GFP}$* mutant also displayed larval arrest at L2 (*Figure 8B,C*) and prominent cytoplasmic GFP signal instead of cell membrane or nuclear signals (*Figure 8D–F*; *Figure 8—figure supplement 1*), supporting that that the extracellular region of MYRF is functionally critical.

Using a similar strategy, we analyzed an in-frame deletion allele *myrf-2$^{1-728\ GFP}$* and found that the extracellular region of MYRF-2 was necessary for MYRF-2's cell membrane localization because MYRF-2$^{1-728}$ was primarily detected in cytoplasm (*Figure 8—figure supplement 2*). These data

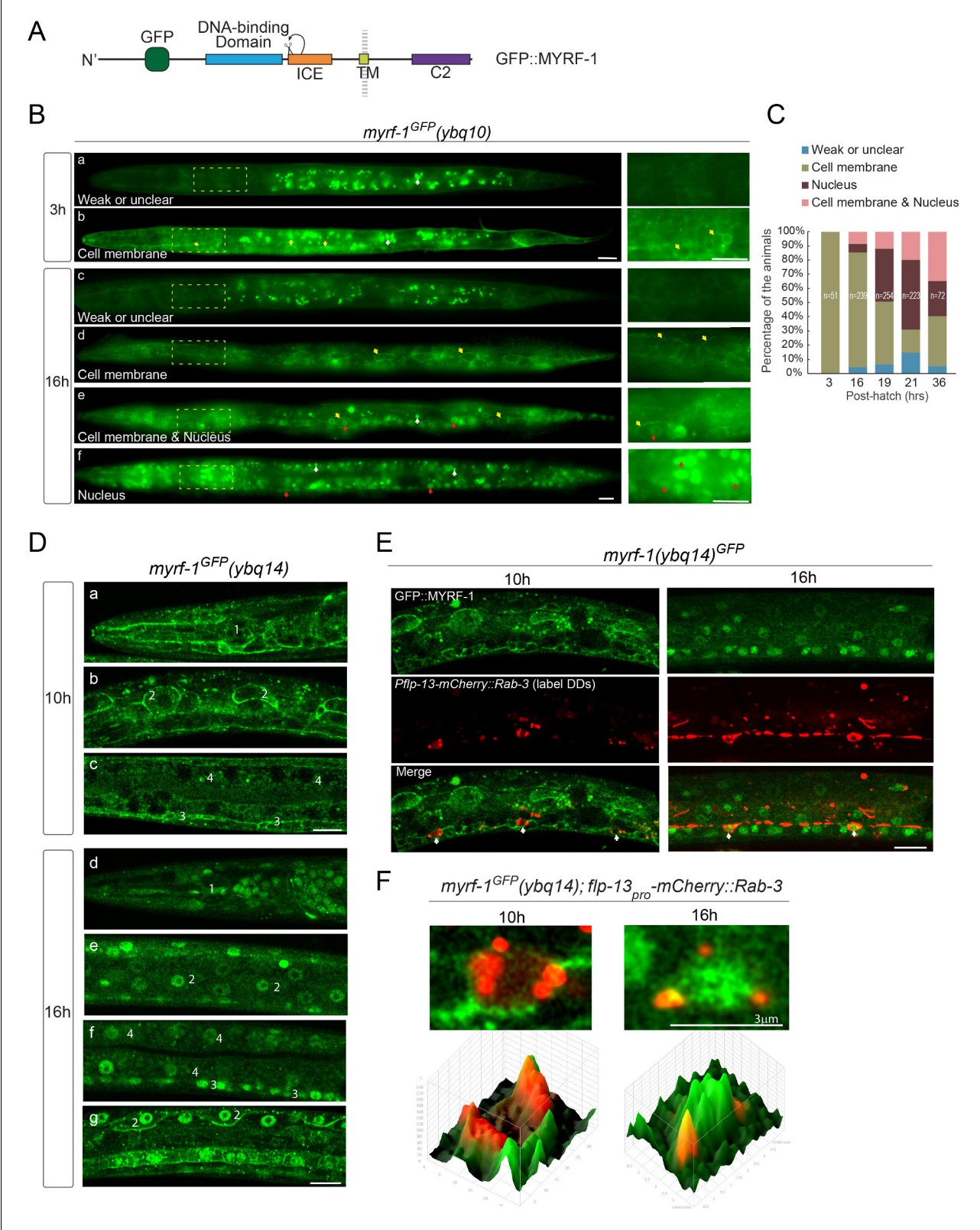

**Figure 4.** MYRF localizes on the cell membrane and is cleaved at specific stages. (**A**) Illustration of functional domains of MYRF. GFP is inserted after Ala171 within the N-MYRF. (**B**) MYRF-1$^{GFP}$ signals in animals of early and late L1 imaged using wide-field microscope and sCMOS camera. GFP was primarily observed at the cell membrane in *myrf-1$^{GFP}$(ybq10)* (Knock in) animals at three post-hatch hours (**b**). Three types of GFP patterns were observed within a population of the animals at 16 post-hatch hours, that is primarily on the cell membrane (**d**), primarily in the nucleus (**f**), and in both

*Figure 4 continued on next page*

*Figure 4 continued*

(e). A small population of animals showed weak, unclear, or inconsistent signals throughout the body (**a, c**). The panels on the right show the framed areas in a-f (dotted lines) with higher magnification. White arrow, auto-fluorescence. Yellow arrow, GFP at the cell membrane. Red arrow, GFP in the nucleus. Scale bar, 10 μm. (**C**) Quantification of animals showing particular pattern of MYRF-1$^{GFP}$ (as in (**B**)) at various stages. (**D**) Single optical-plane images of MYRF-1$^{GFP}$(*ybq14*) at 10, 16 post-hatch hours acquired by Airyscan confocal microscopy. *ybq14* is made by knocking in 3xFlag at C' of *myrf-1* in *ybq10* background (**Meng et al., 2017**). *ybq10* and *ybq14* are of no difference in MYRF-1$^{GFP}$ signals. All sections are sagittal. a, d, head region. b, e, g, lateral (left-right) section of the middle (head-tail) segment of animal. c, f, medial (left-right) section of the middle (head-tail) segment of animal. a, b, c, GFP at the cell membrane. d, e, f, GFP in the nucleus. g, GFP at the cell membrane and in the nucleus. Number 1–4 labels selective anatomical structures and cell types, including the position of metacorpus –the middle part of pharynx (1), seam cell (2), ventral nerve cord (3), intestine (4). Scale bar, 10 μm. (**E**) Colocalization of MYRF-1$^{GFP}$ and DD neuron marker. The animals *myrf-1*$^{GFP}$(*ybq14*); *Pflp-13-mCherry::rab-3*(*ybqls1*) at 10, 16 post-hatch hours were imaged by Airyscan confocal microscopy. The animals carry *glo-1(zu391)* to reduce auto-fluorescence. RAB-3 is localized in cytoplasm. White arrow, soma of DDs. Scale bar, 10 μm. (**F**) Shown are the magnified image of DD soma from the colabeling experiment (**E**), and 3-D intensity plots of green and red signals in the upper image.

The online version of this article includes the following figure supplement(s) for figure 4:

**Figure supplement 1.** MYRF-2 is localized at cell membrane.

## *myrf-1*$^{1-700}$ mutant exhibits precocious, yet discordant synaptic rewiring

While the loss of extracellular domains of MYRF resulted in larval arrest, we examined if such aberrations in MYRF led to defective synaptic rewiring in DDs. Remarkably, the two deletion alleles of *myrf-1* displayed opposite synaptic rewiring abnormalities. *myrf-1*$^{1-790\ GFP}$ allele, in which a part of the extracellular domain of MYRF was removed, exhibited blocked synaptic rewiring under *myrf-2* null background (**Figure 6**; **Figure 6—figure supplements 1–12**), a phenotype similarly observed in *myrf-1(ybq6)*. In contrast, *myrf-1*$^{1-700\ GFP}$ allele, in which the whole extracellular region of MYRF is lost, exhibited precocious synaptic rewiring, that is clearly detectable synaptic puncta formed in the dorsal processes of DDs at 12, 14 post-hatch hours, time points by which synaptic rewiring has not occurred yet in wild type animals (**Figure 6**; **Figure 6—figure supplements 1–12**). The precocious synaptic rewiring in *myrf-1*$^{1-700\ GFP}$ allele also occurred in *myrf-2(0)* background. Notably, the precocious dorsal synapses in *myrf-1*$^{1-700\ GFP}$ mutants were not uniformly distributed along the dorsal cord, that is in some DDs the remodeling occurred while in the rest the remodeling did not, even though they were in the same animal, implying that the observed precocity in *myrf-1*$^{1-700\ GFP}$ mutants was not a strictly regulated event.

To further define the aberrant timing of synaptic rewiring for *myrf-1*$^{1-700\ GFP}$ mutants, we analyzed the M cell division, which undergoes larval stage-specific lineage progression (**Harfe et al., 1998**). We found that *myrf-1*$^{1-700\ GFP}$ caused an advancement in M-cell lineage progression at 12, 14, and 16 post-hatch hours (**Figure 9A**). Curiously, this advancement was eliminated when *myrf-2* was simultaneously mutated (as a putative indel null); even more so, M-cell division was delayed in *myrf-1*$^{1-700\ GFP}$; *myrf-2(0)* double mutants (**Figure 9A**). By dual-labeling M-cell progenitors and synapses of DDs in the same individuals, we determined that the synaptic rewiring was normally initiated when the M-cell underwent the 4th M-cell division (8 + cell); this coordination of timing was disrupted in *myrf-1*$^{1-700\ GFP}$ mutants because many animals had displayed rewired dorsal synapses while only containing less than 4 M-cell progeny cells (**Figure 9B**). This discordance was even more evident in *myrf-1*$^{1-700\ GFP}$; *myrf-2(0)* double mutants because many animals exhibited precocious synaptic rewiring while the M-cell division was delayed (**Figure 9B**). Together, our analyses suggest that the truncate MYRF$^{1-700\ GFP}$, with the whole extracellular domain removed, confers aberrant activities that may accelerate or obstruct development, and importantly, such abnormal progression lacks the inter-tissue coordination characteristic of normal development.

The apparent phenotypic discrepancy between *myrf-1*$^{1-700\ GFP}$ and *myrf-1*$^{1-790\ GFP}$ alleles motivated us to examine localization of the two MYRF variant proteins in more details. When compared the signal patterns of MYRF-1$^{1-700\ GFP}$ and MYRF-1$^{1-790\ GFP}$, we indeed noticed the difference in at least two aspects. The first was that MYRF-1$^{1-700\ GFP}$ signals were consistently brighter than MYRF-1$^{1-790\ GFP}$ signals although they were both predominantly cytoplasmic (**Figure 8G**; **Figure 8—figure supplement 1**). A more striking feature was that the MYRF-1$^{1-700\ GFP}$ signals were observed in nuclei

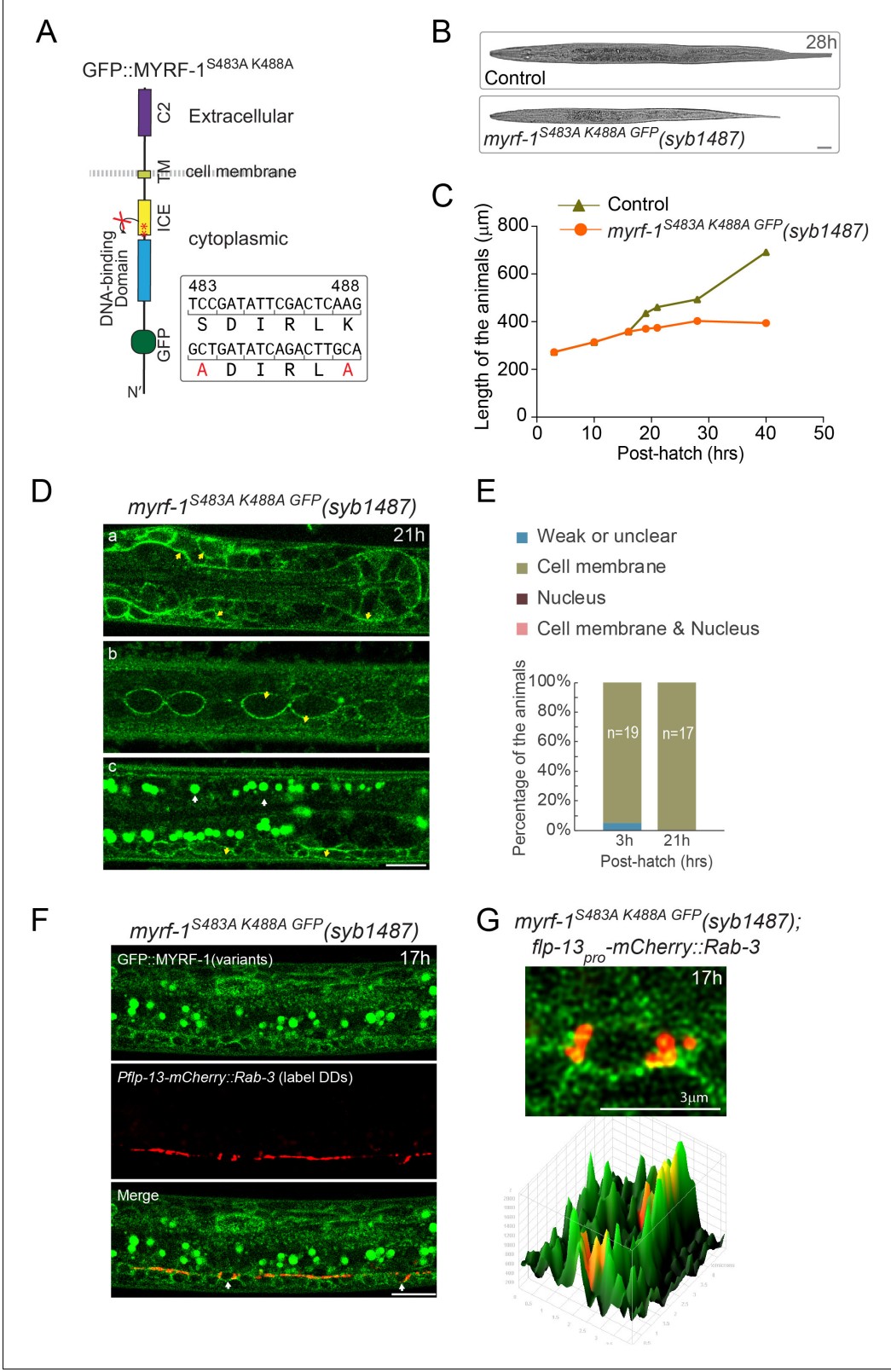

**Figure 5.** Loss-of-cleavage variant of MYRF stays on cell membrane. (**A**) Illustration of loss-of-cleavage MYRF-1 mutant protein encoded by *myrf-1*[S483A K488A GFP](*syb1487*) allele. With the two critical catalytic residues of ICE domain mutated, MYRF-1 cannot be cleaved, and GFP signals are expected to remain at cell membrane. (**B**) DIC images of *myrf-1*[S483A K488A GFP](*syb1487*) and control *myrf-1(syb1487)/mIn1* animals. At 28 post-hatch hours, the

*Figure 5 continued on next page*

*Figure 5 continued*

mutants were alive but shorter because they arrested at earlier stages. Scale bar, 20 μm. (C) Body length of *myrf-1*$^{S483A\ K488A\ GFP}$*(syb1487)* and control animals *myrf-1(syb1487)/mln1* was quantified and shown as Mean. n = 15. (D) GFP signals in different body regions of *myrf-1*$^{S483A\ K488A\ GFP}$*(syb1487)* animals. GFP was primarily detected at the cell membrane in the mutant at 21 hr. All optical sections are sagittal. a, head region. b, lateral (left-right) section of trunk region showing seam cells (yellow arrows). c, medial (left-right) section of trunk region showing ventral cord neurons (yellow arrows). White arrow, auto-fluorescence. Yellow arrow, GFP at the cell membrane. Images are single optical slices by Airyscan confocal microscopy. Scale bar, 10 μm. (E) Quantification of animals showing particular pattern of MYRF-1$^{S483A\ K488A\ GFP}$ (as in (D)) at indicated stages. (F) Colocalization of MYRF-1$^{S483A\ K488A\ GFP}$ and DD neuron marker. The animals *myrf-1*$^{S483A\ K488A\ GFP}$*(syb1487); Pflp-13-mCherry::rab-3(ybqls1)* at 17 post-hatch hours were imaged by confocal microscopy. RAB-3 is localized in cytoplasm. White arrow, soma of DDs. Scale bar, 10 μm. (G) Shown are the magnified image of DD soma from the colabeling experiment (F), and 3-D intensity plots of green and red signals in the upper image.

---

of many cells, which is especially evident in big nuclei such as those for seam cells, epidermal cells, and intestinal cells. In contrast, similar nuclear GFP signals were not observed in *myrf-1*$^{1-790\ GFP}$ mutants. These observations were supported by analyzing the GFP intensity distribution along a line across a single seam cell or ventral cord neuron (*Figure 8G*). We found that MYRF-1$^{1-700\ GFP}$ signals exhibited an elevation of intensity in nuclear regions, even though the pattern across a population of animals lacked regularity. In contrast, MYRF-1$^{1-790\ GFP}$ signals displayed uniformly saddle-shaped pattern suggesting the absence of nuclear signals. The relatively weak, yet detectable nuclear signals of MYRF-1$^{1-700\ GFP}$ suggested that it could be processed on ER to release the N-MYRF-1$^{GFP}$(1-482); however, the process was incomplete because prominent GFP signals remained in cytoplasm (*Figure 8F,G*). Since the nuclear signals of MYRF-1$^{1-700\ GFP}$ were observed at stages earlier than late L1, for example 10 post-hatch hours, we deduced that the cleavage of MYRF-1$^{1-700\ GFP}$ on ER was constitutively active. This situation was distinct from what was for wild type MYRF-1, because cleavage of MYRF-1 appeared to be inhibited until late L1. Thus, the differential localization of MYRF-1$^{1-700\ GFP}$ and MYRF-1$^{1-790\ GFP}$ was consistent with the contrasting synaptic rewiring defects in the corresponding mutants.

## PAN-1 is required for MYRF's cell-membrane and nuclear localization

Since PAN-1 and MYRF both localize on cell membrane and interact with each other, we asked how the loss of one factor would affect the localization of the other (*Figure 10*). The distribution of PAN-1$^{GFP}$ in *myrf-1(ju1121)* mutants was comparable to that in wild type animals (*Figure 10E*; *Figure 10— figure supplement 1*), indicating that *myrf-1* is unlikely to regulate PAN-1's stability or trafficking. In stark contrast, MYRF-1$^{GFP}$ or MYRF-2$^{GFP}$ signal, both on the membrane and in the nucleus, was barely detected in *pan-1* mutants at all stages examined (*Figure 10A*; *Figure 10—figure supplement 1A*), indicating that PAN-1 critically regulates MYRF localization. The loss of MYRF$^{GFP}$ in *pan-1 (0)* was not due to failed transcription of *myrf* genes, as *myrf-1* and *myrf-2* transcript levels were not significantly changed in *pan-1(0)* mutants (*Figure 10B*).

One explanation for the disappearance of MYRF in *pan-1* mutants is that MYRF protein undergoes degradation in the absence of PAN-1. We investigated this possibility by blocking the ubiquitin-mediated proteasome system (*Kipreos, 2005*) and testing if MYRF protein could be restored in *pan-1* mutants (*Figure 10C*). When critical regulators of the proteasome system were knocked down in *pan-1(0)* animals, MYRF$^{GFP}$ signals re-emerged in many cells, indicating that MYRF proteins are subject to proteasome degradation in the absence of PAN-1. Importantly, restored MYRF$^{GFP}$ in *pan-1* mutants was not localized to the cell membrane or nucleus; rather, the signal appeared in ER-like structures in the cytoplasm, suggesting that interaction between PAN-1 and MYRF initiates when they are in ER. In conclusion, PAN-1 is necessary for normal localization and function of MYRF.

We asked if exogenously expressed *pan-1* can restore the cell-membrane localization of MYRF in *pan-1* mutants. To this end, we used a seam cell-specific promoter to drive *pan-1* expression (*Figure 10D*) and observed restored nuclear MYRF-1$^{GFP}$ signal in seam cells of *pan-1(0)* animals, indicating that MYRF had been normally localized and processed in those cells. This result also supports a cell-autonomous function for *pan-1*.

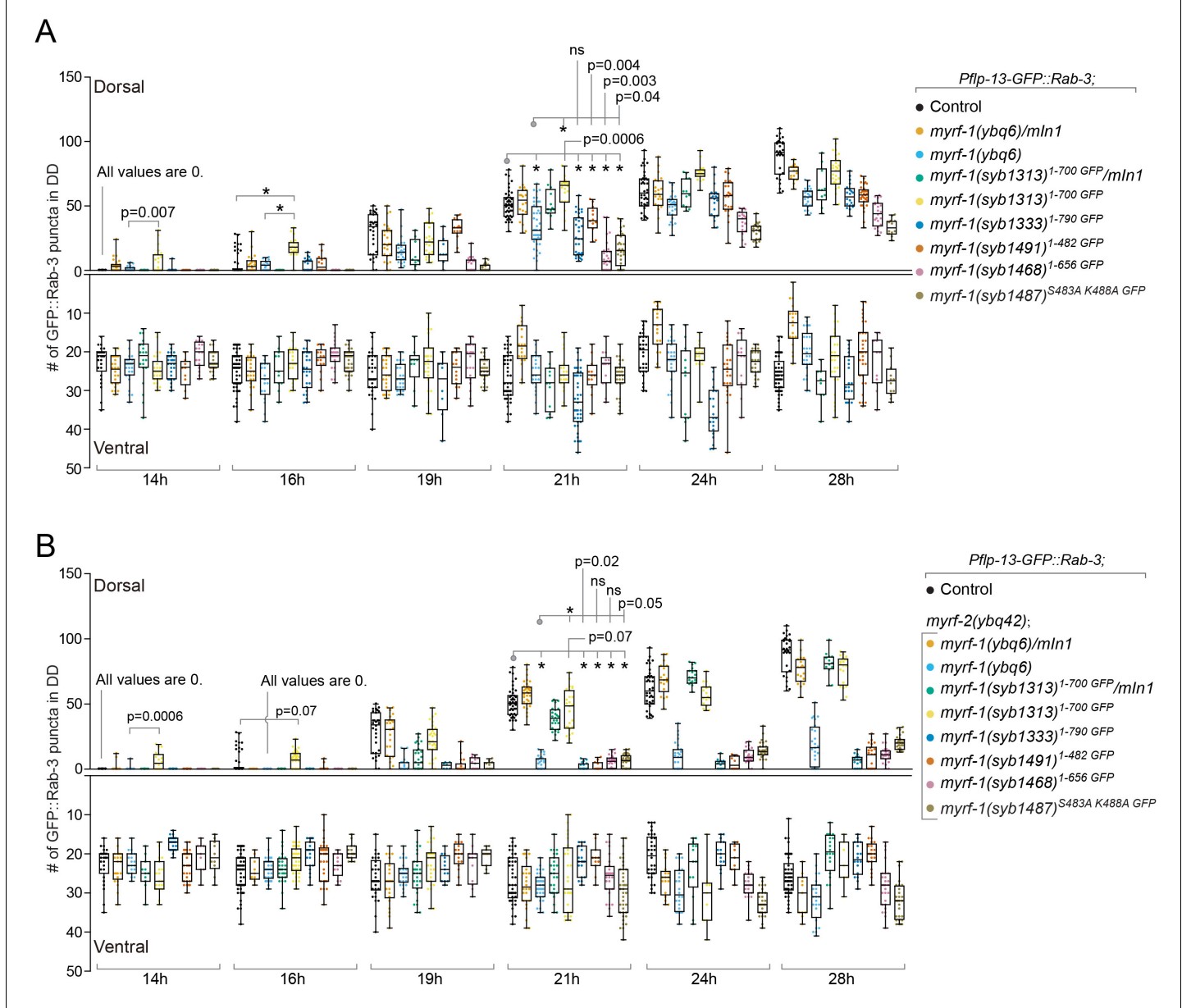

**Figure 6.** Analysis of synaptic rewiring in *myrf-1* mutants. Quantifications of the ventral and dorsal synapses in *myrf-1* single mutants (**A**) and *myrf-1; myrf-2* double mutants (**B**). Wild-type control (marker only) is *flp-13_pro_-GFP::rab-3(ybqIs47)*. The allele genotypes of each mutant are indicated in the figure. All mutants carry *ybqIs47* marker. The number of synapses in DDs were counted and shown as Mean ± SEM (t test. ns, p>0.05. *, p<0.0001.). Indicated time points are post-hatch hours. p-Values for comparison between all genotypes can be found in *Supplementary file 4*.

The online version of this article includes the following figure supplement(s) for figure 6:

**Figure supplement 1.** Synaptic rewiring in *myrf-1* mutants at 14 hr.
**Figure supplement 2.** Synaptic rewiring in *myrf-1* mutants at 16 hr.
**Figure supplement 3.** Synaptic rewiring in *myrf-1* mutants at 19 hr.
**Figure supplement 4.** Synaptic rewiring in *myrf-1* mutants at 21 hr.
**Figure supplement 5.** Synaptic rewiring in *myrf-1* mutants at 24 hr.
**Figure supplement 6.** Synaptic rewiring in *myrf-1* mutants at 28 hr.
**Figure supplement 7.** Synaptic rewiring in *myrf-1; myrf-2* double mutants at 14 hr.
**Figure supplement 8.** Synaptic rewiring in *myrf-1; myrf-2* double mutants at 16 hr.
**Figure supplement 9.** Synaptic rewiring in *myrf-1; myrf-2* double mutants at 19 hr.
**Figure supplement 10.** Synaptic rewiring in *myrf-1; myrf-2* double mutants at 21 hr.
**Figure supplement 11.** Synaptic rewiring in *myrf-1; myrf-2* double mutants at 24 hr.

*Figure 6 continued on next page*

*Figure 6 continued*

**Figure supplement 12.** Synaptic rewiring in *myrf-1; myrf-2* double mutants at 28 hr.

The propensity of MYRF being degraded in the absence of PAN-1 implies that the non-cytoplasmic region of MYRF is propone to degradation. Since the whole extracellular region of MYRF was removed in MYRF-1$^{1-700\ GFP}$, the mutant protein was predicted to be able to evade the degradation surveillance even when PAN-1 was unavailable. The prediction proved to be correct because MYRF-1$^{1-700\ GFP}$ signals remained strong in *pan-1(0)* mutants (**Figure 10F**). In contrast, the signals of MYRF-1$^{1-790\ GFP}$, in which part of extracellular region was removed, were undetectable in *pan-1(0)*, resembling wild-type MYRF in *pan-1(0)*. Therefore, PAN-1 is essential for stabilizing MYRF and trafficking it to cell membrane.

## MYRF-PAN-1 interaction is important for synaptic rewiring

To further test if MYRF functionally interacts with PAN-1 at the cell membrane, we aimed to disrupt their interaction by overexpressing the extracellular region of MYRF-1 and analyzed how the perturbation affected MYRF's activity (**Figure 11A**). To this end, we generated two fusion constructs containing the extracellular region of MYRF-1 (MYRF-1$^{681-931}$): one fused to TMD of NEP-2, a type II transmembrane metallopeptidase Neprilysin (**Yamada et al., 2010**), and the other fused to the SP of MIG-17, a secreted metallopeptidase in the ADAMTS family (**Nishiwaki, 2000**). Overexpression of either fusion protein (TMD::MYRF-1$^{681-931}$ and SP:: MYRF-1$^{681-931}$) in DD neurons blocked synaptic remodeling, supporting that the interaction between PAN-1 and MYRF is important for MYRF activity.

Since the interaction of PAN-1 and MYRF likely begins in ER, we seek a way to trap MYRF in ER so that MYRF does not traffick to cell membrane. We expressed a modified PAN-1 without transmembrane domain and fused with KDEL motif, the ER retention signal, in DDs (**Figure 11B–D**). This altered PAN-1 protein was expected to be retained inside the ER lumen, and consequently blockade MYRF trafficking out of ER through binding MYRF. The ER-retained PAN-1 indeed impeded the rewiring processes drastically, as the dorsal synapses were greatly reduced and most of the juvenile ventral synapses were sustained by 28 post-hatch hours. When ER-retained PAN-1 was expressed in DDs as well as seam cells of animals carrying *myrf-1$^{GFP}$*, we observed much brighter MYRF$^{GFP}$ signals in target cells than in neighbor cells, and importantly, the signals were primarily cytoplasmic showing a pattern similar to ER structures. These results support that PAN-1 promotes MYRF's transport to cell membrane, and that PAN-1 interacting with MYRF in ER is insufficient for MYRF's cleavage.

## N-terminal MYRF alone is insufficient for its activity

Our data collectively indicate that trafficking to the cell membrane is a prerequisite step for proteolytic processing of MYRF. We asked if expression of free N-MYRF could circumvent the need for full-length MYRF to traffic to the cell membrane, and recapitulate MYRF function (**Figure 12**). We generated an in-frame deletion allele of *myrf-1$^{1-482\ GFP}$* that removed the entire carboxyl region from the cleavage point to the end of MYRF. This mutant MYRF-1$^{1-482\ GFP}$ concentrated in the nucleus throughout the body (**Figure 12**; **Figure 12—figure supplement 1**), consistent with N-MYRF possessing a nuclear localization signal and nuclear function (**Li et al., 2013**; **Meng et al., 2017**). However, despite MYRF-1$^{1-482\ GFP}$ nuclear localization, the mutants arrested at L2, similar to the *myrf-1* null allele, indicating that N-MYRF alone cannot substitute for full-length MYRF.

Studies of mammalian MYRFs have demonstrated that a trimeric form of N-MYRF exhibits higher transcriptional activity than the monomer (**Li et al., 2013**). The presence of the ICA region is necessary and sufficient for mammalian MYRF to form a trimer and undergo cleavage (**Li et al., 2013**). It is possible that MYRF-1$^{1-482\ GFP}$ was insufficient because N-MYRF alone cannot form a trimer. To test this possibility, we generated an in-frame deletion allele of *myrf-1$^{1-656\ GFP}$*, which included the ICA region that enables MYRF trimerization (**Figure 12**). In this mutant, GFP-MYRF-1$^{1-656}$ accumulated in nuclei throughout the body at all viable stages, but the animals displayed larval arrest similar to that in *myrf-1* null animals.

The synaptic rewiring in both *myrf-1$^{1-482\ GFP}$* and *myrf-1$^{1-656\ GFP}$* showed deficiency similar to *myrf-1(ybq6)* (**Figure 6**; **Figure 6—figure supplements 1–12**) Therefore, the analysis of the two

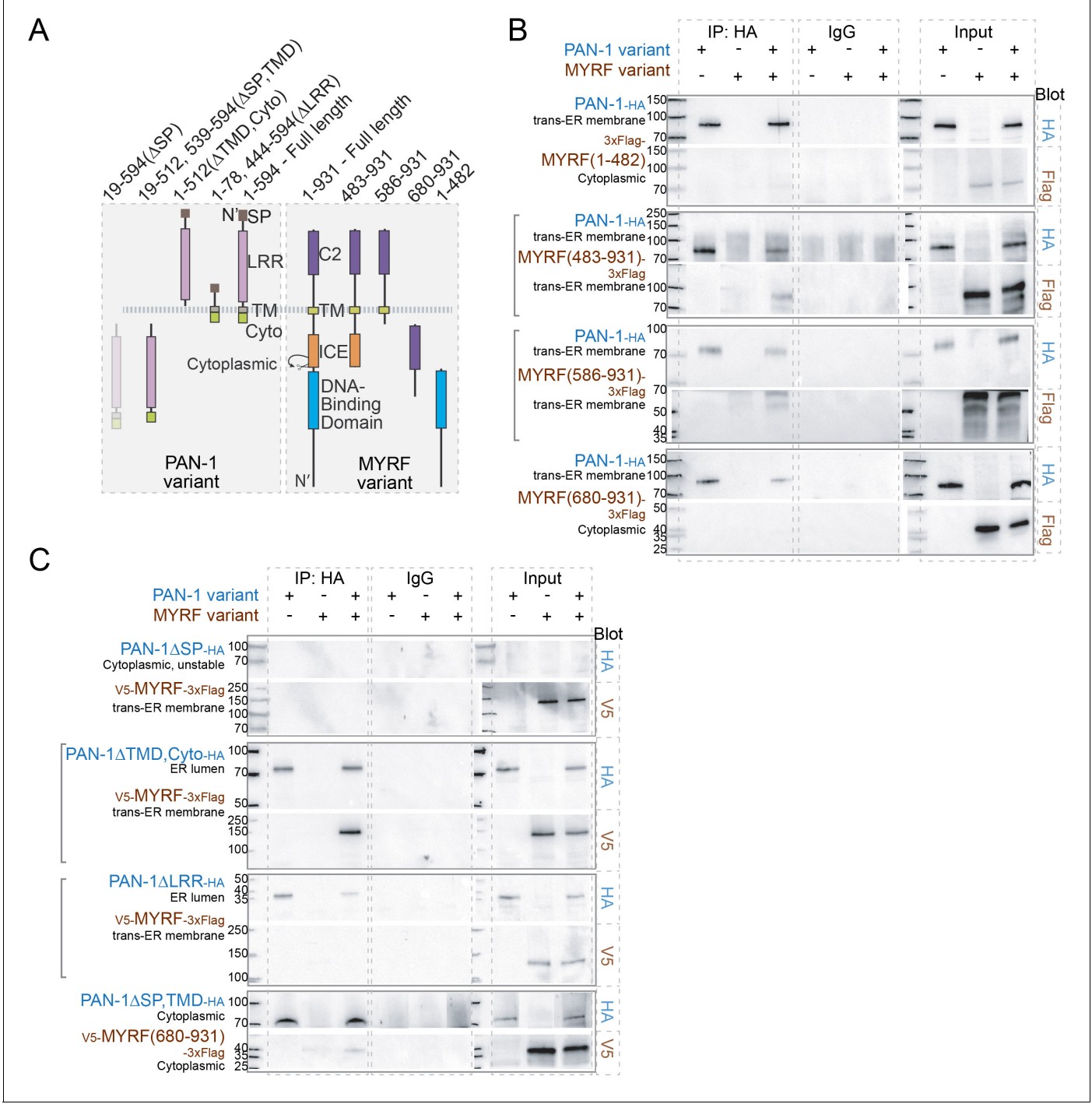

**Figure 7.** MYRF and PAN-1 interact through extracellular domains in HEK293T cells. (**A**) Illustration of PAN-1 and MYRF protein variants (used in B and C) that were expressed in HEK293T cells, showing predicted topology on membrane. (**B**) PAN-1 interacted with the non-cytoplasmic region of MYRF-1. Co-IP of PAN-1 and MYRF-1 variants from the lysate of HEK293T cells was performed. Rabbit anti-HA was used in co-IP, and the target bands in western blot were detected using murine anti-HA, murine anti-V5 or murine anti-Flag. The subcellular compartment for each expressed protein has been indicated in the figure. Square brackets indicate the pairs that showed positive interaction. (**C**) The LRR of PAN-1 interacted with the non-cytoplasmic region of MYRF-1. Co-immunoprecipitation of PAN-1 variants and MYRF form the lysate of HEK293T cells was performed similarly to (**B**).

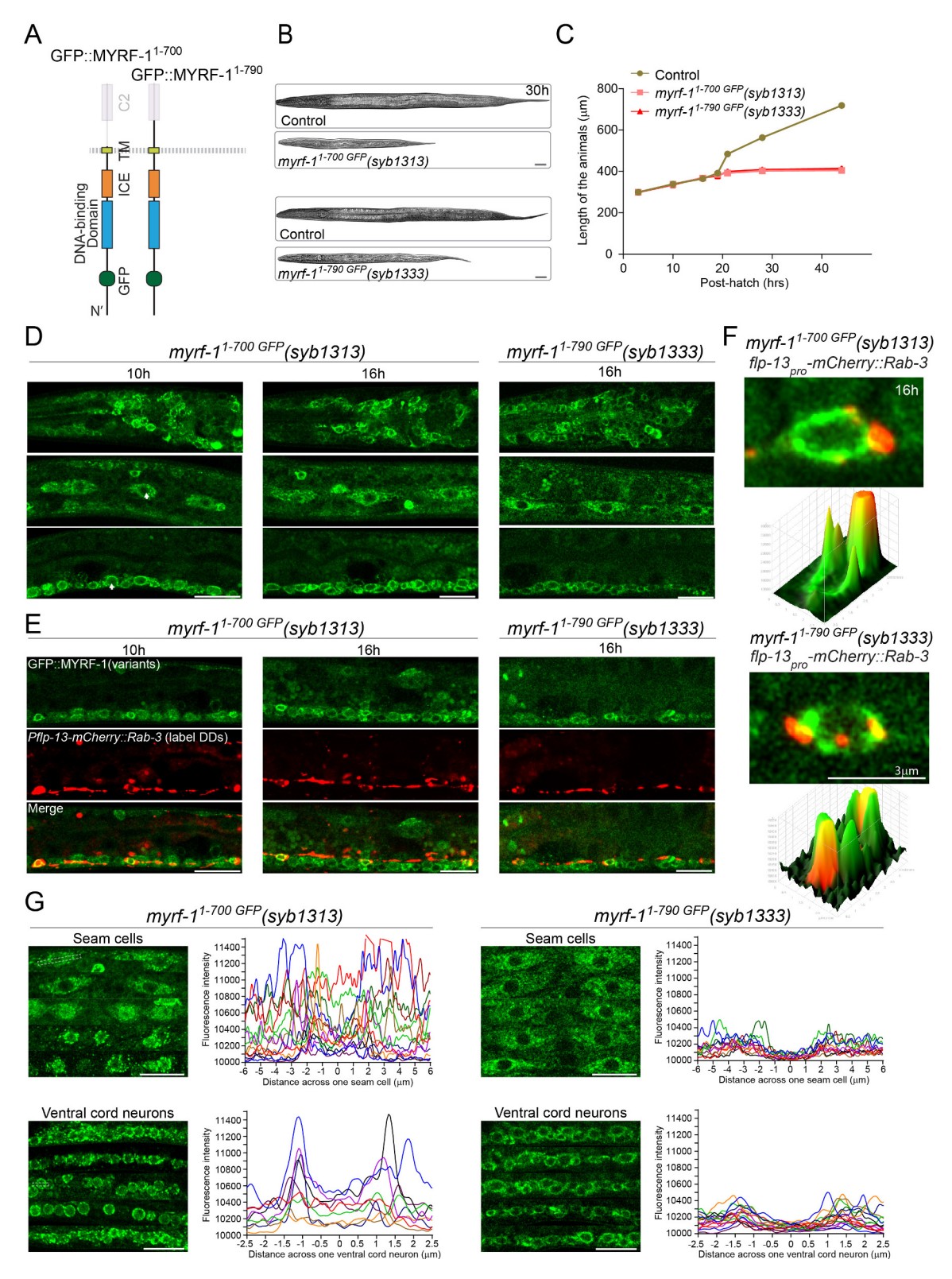

**Figure 8.** Extracellular region is required for MYRF's cell-membrane localization and function. (**A**) Illustration of truncate MYRF-1 proteins encoded by *myrf-1*[1-700 GFP](syb1313) and *myrf-1*[1-790 GFP](syb1333), with in-frame deletion of the whole and the part of the extracellular region of MYRF-1, respectively. (**B**) DIC images of *myrf-1*[1-700 GFP](syb1313) and control *myrf-1(syb1313)/mln1* animals (top); *myrf-1*[1-790 GFP](syb1333) and control *myrf-1 (syb1333)/mln1* animals (bottom). At 30 post-hatch hours, the mutants were shorter because they arrested at earlier stages. Scale bar, 20 µm. (**C**) Body
*Figure 8 continued on next page*

*Figure 8 continued*

length of *myrf-1*[1-700 GFP](*syb1313*), *myrf-1*[1-790 GFP](*syb1333*), and control animals *myrf-1(syb1313)/mln1* was measured and shown as Mean, n = 15. (D) GFP signals in different body regions of *myrf-1*[1-700 GFP](*syb1313*) and *myrf-1*[1-790 GFP](*syb1333*) mutants. GFP signals were primarily localized in the cytoplasm in both mutants. Weak GFP signals were also detected in the nuclei of *myrf-1*[1-700 GFP](*syb1313*) (white arrows), but not in those of *myrf-1*[1-790 GFP](*syb1333*). Little cell membrane signals were observed in either mutant. Images are single sagittal optical slices obtained by AiryScan confocal microscopy. Top row, head regions. Middle row, lateral (left-right) section of middle (head-tail) segment with a focus on seam cells. Bottom row, medial (left-right) section of middle (head-tail) segment with a focus on ventral nerve cord. Scale bar, 10 μm. (E) Colabeling MYRF-1[1-700 GFP](*syb1313*) and MYRF-1[1-790 GFP](*syb1333*) with DD neuron marker *Pflp-13-mCherry::rab-3(ybqls1)* at 10, 16 post-hatch hours. The animals carry *glo-1(zu391)* to reduce auto-fluorescence. Shown are single optical slices by Airyscan confocal microscopy. RAB-3 is localized in cytoplasm. White arrow, soma of DDs. Scale bar, 10 μm. (F) Shown are the magnified image of DD soma from the colabeling experiment (E), and 3-D intensity plots of green and red signals in the upper image. (G) Analysis of GFP intensity across individual seam cells and ventral cord neurons of *myrf-1*[1-700 GFP](*syb1313*) and *myrf-1*[1-790 GFP](*syb1333*) animals. Images excerpts from five independent animals are vertically tiled. Each graph line represents average GFP intensity along a bar ROI across single seam cell (or ventral cord neuron) from independent individual animals. X-axis, position on the bar ROI. 0 on X-axis denotes the center of the cell analyzed. An example of ROI was drawn in the figure as a white dot line square. Scale bar, 10 μm.

The online version of this article includes the following figure supplement(s) for figure 8:

**Figure supplement 1.** Comparing the signal intensity in MYRF-1[1-700 GFP](*syb1313*) and MYRF-1[1-790 GFP](*syb1333*).

**Figure supplement 2.** Extracellular domain of MYRF-2 is required for its cell-membrane localization.

N-MYRF-only mutants support that MYRF membrane-association is indispensable for MYRF's function (*Figure 13*).

## Discussion

Remodeling of neural circuits is driven by intricate interplays between inductive signals and transcriptional programs. We have identified an LRR-TM protein PAN-1 as an essential factor for promoting synaptic remodeling in *C. elegans*. PAN-1 interacts with MYRF on the cell membrane, and the interaction is required for MYRF's cell membrane localization and nuclear translocation (*Figure 13*). We find that MYRF needs to be trafficked onto the cell membrane in order for its N-terminal region to be cleaved and released. The release of N-MYRF from the cell membrane is developmentally regulated, and the timing of release coincides with when MYRF activity is critically required.

Our results, for the first time, show that MYRF is subcellularly localized on the cell membrane in animal development. Previous characterization of MYRF by immunostaining transfected cells or optic nerves of adult mice (for oligodendrocytes) concluded that the full-length MYRF resides on ER before undergoing constitutive self-cleavage; but none of the studies has described if MYRF could be detected on the cell membrane (*Bujalka et al., 2013*; *Emery et al., 2009*; *Li et al., 2013*). Our observation about MYRF's localization on cell membrane raises the question if mammalian MYRF also localizes to the cell membrane prior to cleavage. A thorough examination of MYRF localization in developing mouse tissues would be informative.

Our results demonstrate that PAN-1 is required for proper function of MYRF. One possibility is PAN-1 is constitutively permissive for MYRF function, for example PAN-1 stabilizes the extracellular domain of MYRF; however, aberrant processing and function of MYRF[1-700 GFP] indicate that the ectodomain of MYRF is indispensable, further suggesting a regulatory role for PAN-1. The extracellular domain of PAN-1 is highly homologous to the ectodomain of human Toll-like receptor 5 (TLR5), a member of the toll-like receptor (TLR) family (*Dolan et al., 2007*; *Gissendanner and Kelley, 2013*). TLR5 is known to recognize bacterial flagellin to initiate an intracellular signaling cascade in innate immunity response (*Hayashi et al., 2001*). Although PAN-1 lacks the intracellular motifs of TLR5, its extracellular LRR domains may serve as a receptor for an unidentified ligand. Because MYRF cleavage is dependent on trimerization via its ICE region, MYRF interaction with PAN-1 may alter the state of MYRF trimerization, which affects cleavage.

The extent of regulation of MYRF activity on the membrane remains unknown. One factor, Tmem98, has been identified as a MYRF interactor in both *C. elegans* and mouse (*Huang et al., 2018*). The two proteins' interaction on the membrane inhibits MYRF cleavage in human cell lines. The repressive effect from Tmem98 implies that mammalian MYRF can be regulated on the membrane *in vivo*, while the regulators are not necessarily expressed in cultured cancer cells.

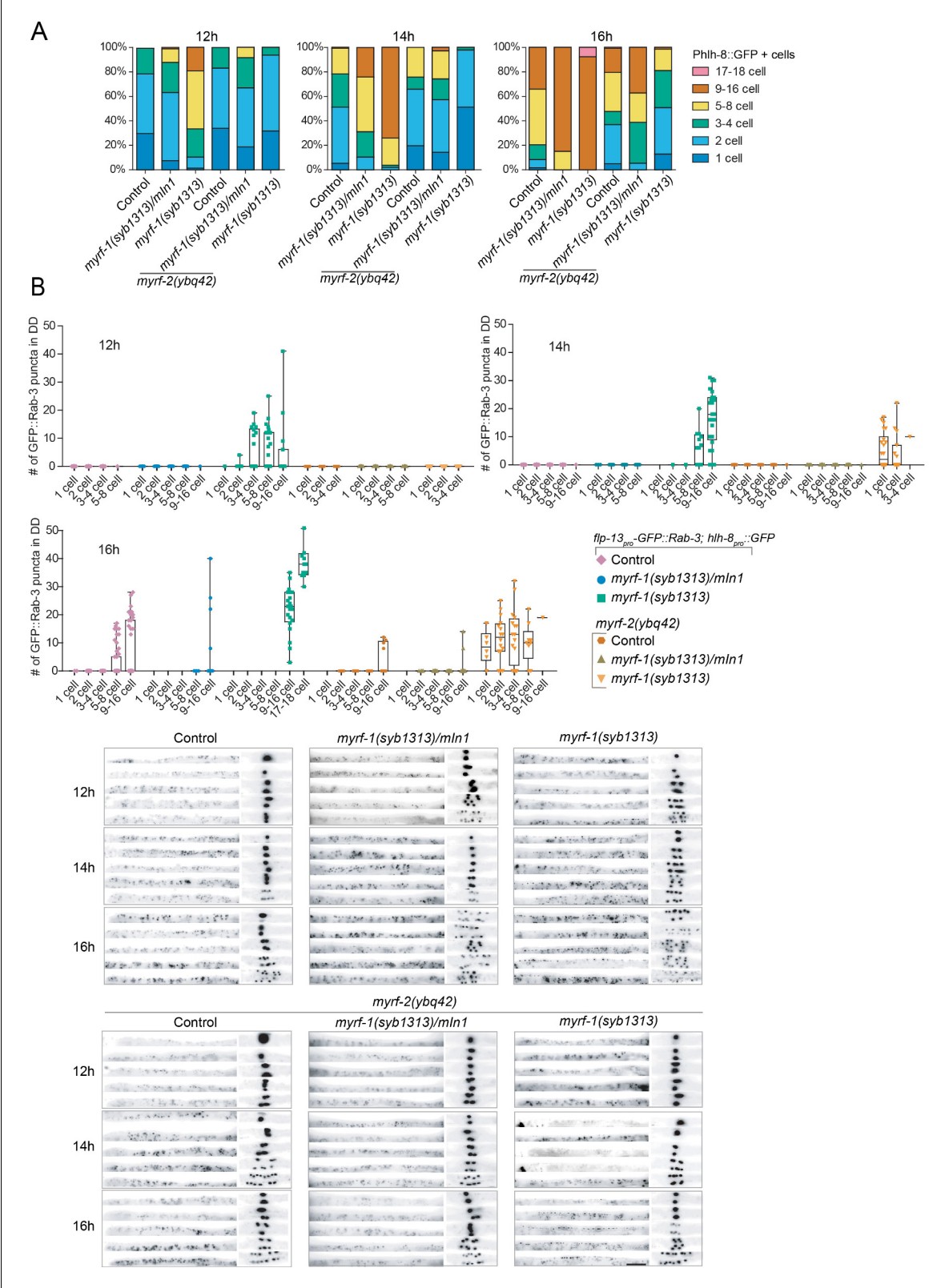

**Figure 9.** *myrf-1^{1-700 GFP}(syb1313)* exhibits precocious, yet discordant synaptic rewiring and M-cell division. (**A**) Percentage of animals showing the particular number of M-cell progenies labeled by *hlh-8_{pro}-GFP(ayIs6)* at 12, 14, 16 post-hatch hours. (**B**) Quantification of dorsal synapse number of DDs in individuals subgrouped by the number of M-cell progenies. The synapses of DDs were labeled by *flp-13_{pro}-GFP::Rab-3(ybqIs47)*. At bottom are representative images showing dorsal cord and M-cell progenies simultaneously. Scale bar, 20 μm.

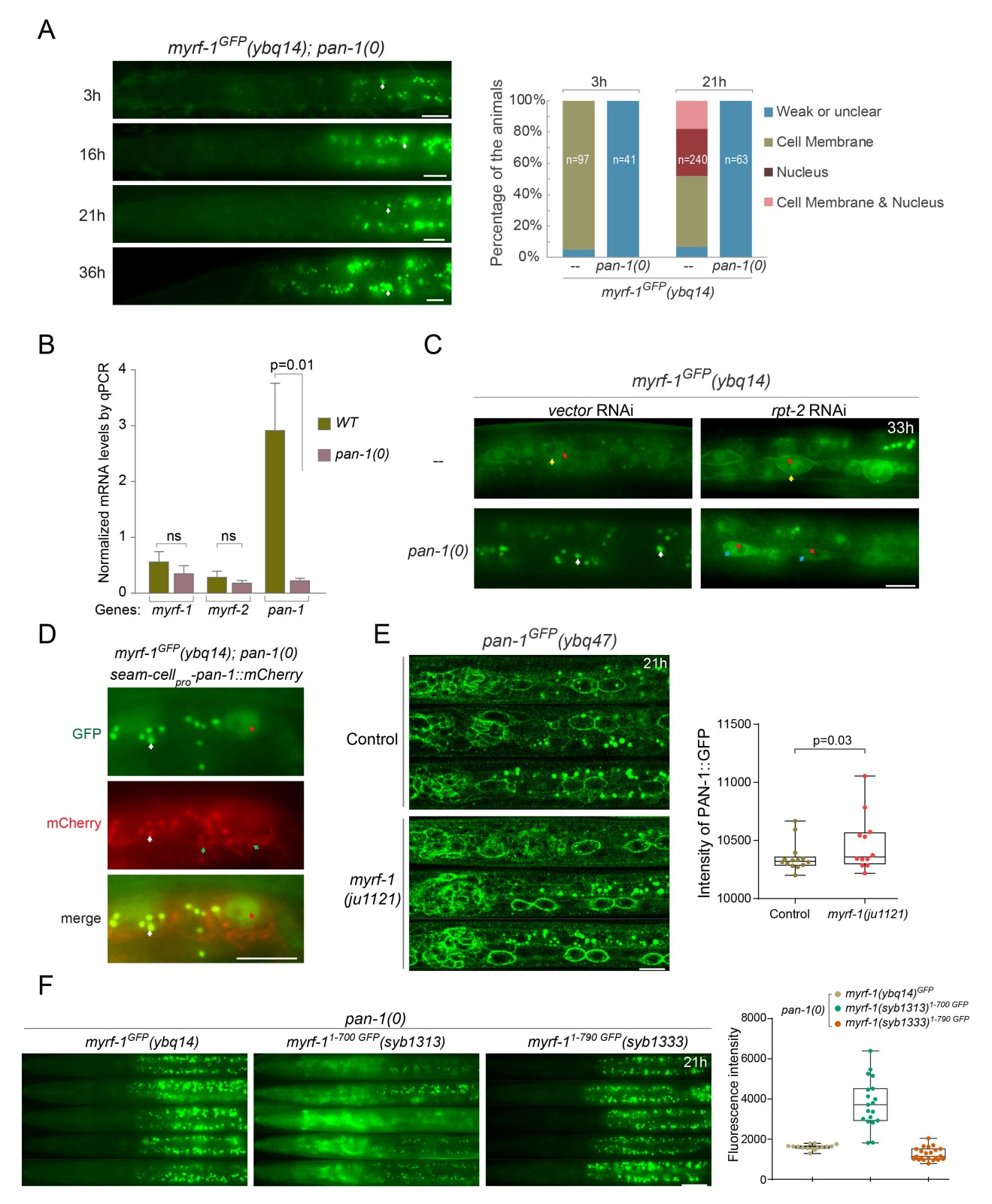

**Figure 10.** PAN-1 is required for MYRF's cell-membrane and nuclear localization. (**A**) Signals of MYRF-1$^{GFP}$(*ybq14*) in *pan-1(gk142)* animals at different post-hatch hours. No clear GFP signals are detected in *pan-1* mutants. White arrow, auto-fluorescence. Images are acquired by wide-field microscopy, showing anterior half of each animal. Scale bar, 10 μm. The graph at right shows the percentage of animals with particular MYRF-1$^{GFP}$ pattern in *pan-1* mutants and control animals at 3, 21 post-hatch hours. (**B**) Transcripts levels of *myrf* and *pan-1* in *pan-1(gk142)* and control animals (only with markers)

*Figure 10 continued on next page*

*Figure 10 continued*

(*unc-25*$_{pro}$-*mCherry::rab-3(juIs236)*; *hlh-8*$_{pro}$-*GFP*(ayIs6)) by qPCR. Mean normalized transcript level of three replicates were shown (Mean ± SEM, t test. ns, p>0.05.). Transcript level of Internal control gene *pmp-3* was used as normalization reference. The animals were 21 post-hatch hours, which were staged with the help of M-cell division marker (*hlh-8*$_{pro}$-*GFP*). (C) Signals of MYRF-1$^{GFP}$(*ybq14*) in *pan-1*(*gk142*) animals under treatment of *rpt-2* RNAi. *rpt-2* is a component of ubiquitin proteasome system. MYRF-1$^{GFP}$ signals are not detected in *pan-1* mutants, but are detected when treated by *rpt-2* RNAi. The GFP signals are observed in cytoplasm (blue arrows) in *pan-1(gk142)*; *rpt RNAi*, in contrast to on the cell membrane (yellow arrows) and in the nucleus (red arrows) in wild-type animals. Scale bar, 10 µm. (D) Signals of MYRF-1$^{GFP}$(*ybq14*) in seam cells of *pan-1(gk142)* mutants can be restored by seam-cell specifically expressed *pan-1* transgene, *SCM*$_{pro}$-*pan-1::mCherry(ybqEx778)*. Red arrow, GFP signals in the nucleus of seam cells. Green arrow, signals of exogenous expressed PAN-1::mCherry in seam cells. White arrow, auto-fluorescence. Scale bar, 10 µm. (E) Signals of PAN-1$^{GFP}$ in control (*pan-1(ybq47)*$^{GFP}$ only) and *myrf-1(ju1121)* mutants, showing anterior half of animal body. Images are single slices acquired by Airyscan. Shown are lateral position of sagittal sections focusing on seam cells. Images excerpts from three independent animals are vertically tiled. Scale bar, 10 µm. The graph at right shows the intensity of PAN-1$^{GFP}$ on cell membrane of seam cells. Each data point represents independent individual animals. Shown are Mean ± SEM (t test). (F) MYRF-1$^{GFP}$(*ybq14*), MYRF-1$^{1-700\ GFP}$(*syb1313*), and MYRF-1$^{1-790\ GFP}$(*syb1333*) signals in *pan-1(gk142)* mutants. Images are captured on wide-field microscope and show the anterior half of animal body. Images excerpts from five independent animals are vertically tiled. Scale bar, 20 µm. The graph at right shows the average intensity of head region (which is free of intestinal auto-fluorescence). Each data point represents individual animal.

The online version of this article includes the following figure supplement(s) for figure 10:

**Figure supplement 1.** *pan-1* is required for cell-membrane localization of MYRF.

Our results provide the first direct evidence that the non-cytoplasmic part of MYRF protein is indispensable for its function. Without it, MYRF failed to traffic to the cell membrane; instead, MYRF was most likely trapped in ER in *C. elegans*. Previous *in vitro* studies demonstrated that truncated human MYRF(1-756), lacking the transmembrane domain and non-cytoplasmic region, was much less competent at promoting oligodendrocyte maturation than full-length MYRF, even though the truncated MYRF(1-756), when overexpressed, can be processed to release N-MYRF in cultured cells (*Li et al., 2013*). The diminished activity of truncated MYRF suggests that the C-terminal region of human MYRF confers additional function.

Two recent human genetics studies identified that haplotypes of *MYRF* C-terminal variants can cause nanophthalmos, a potentially devastating eye condition (*Rossetti et al., 2019*; *Siggs et al., 2019*). The two identified *MYRF* mutations encoded a full-length MYRF peptide, but a reading frame shift caused the final 26 amino acids to be replaced with 30 different amino acids. These findings support the functional importance of the MYRF C-tail region and corroborate with our observation that the MYRF extracellular domain is essential for MYRF's activity.

We previously determined that overexpressing N-MYRF-1$^{1-482}$ in DDs sufficiently accelerated the synaptic rewiring (*Meng et al., 2017*). However, in present study, endogenous expression of N-MYRF (as in *myrf-1*$^{1-482\ GFP}$, *myrf-1*$^{1-656\ GFP}$) appeared to be deficient in MYRF function, even though N-MYRF-1 in all cases concentrated in the nucleus. We reasoned that overexpressed transgene N-MYRF-1 could sufficiently form trimers, a prerequisite for its full transcriptional activity, whereas endogenously expressed N-MYRF-1 could not. To test the hypothesis, we generated two single-copy transgene lines, expressing full-length MYRF-1 and N-MYRF-1, respectively. We found that the full-length MYRF-1 could rescue the defective synaptic rewiring in *myrf-1*; *myrf-2* double mutants, while N-MYRF-1 could not (*Figure 12—figure supplement 2*). This result was consistent with the defective synaptic rewiring in *myrf-1*$^{1-482\ GFP}$ mutants. It remains to be experimentally determined if N-MYRF in *myrf-1*$^{1-482\ GFP}$, *myrf-1*$^{1-656\ GFP}$ indeed fail to form trimer efficiently. It is conceivable that membrane localization may improve thermodynamic conditions for MYRF oligomerization, even though our analysis of MYRF$^{1-700\ GFP}$ indicate that merely situating on ER membrane (without MYRF's ectodomain) is insufficient for MYRF's cleavage.

In cultured cells, human MYRF is found to carry post-translational modifications, for example glycosylation (*Li et al., 2013*). Membrane association may be required for proper modification of MYRF, and the modifications may facilitate nucleation of N-MYRF transcriptional complexes on target promoters. Indeed, sumoylation of MYRF's activator domain promotes its transcriptional activity (*Choi et al., 2018*). It is also possible that specific factors are co-transported with N-MYRF from the cell membrane into the nucleus, thus affecting MYRF's transcription activity. These yet-to-be-defined features may enable essential MYRF function in regulating animal development.

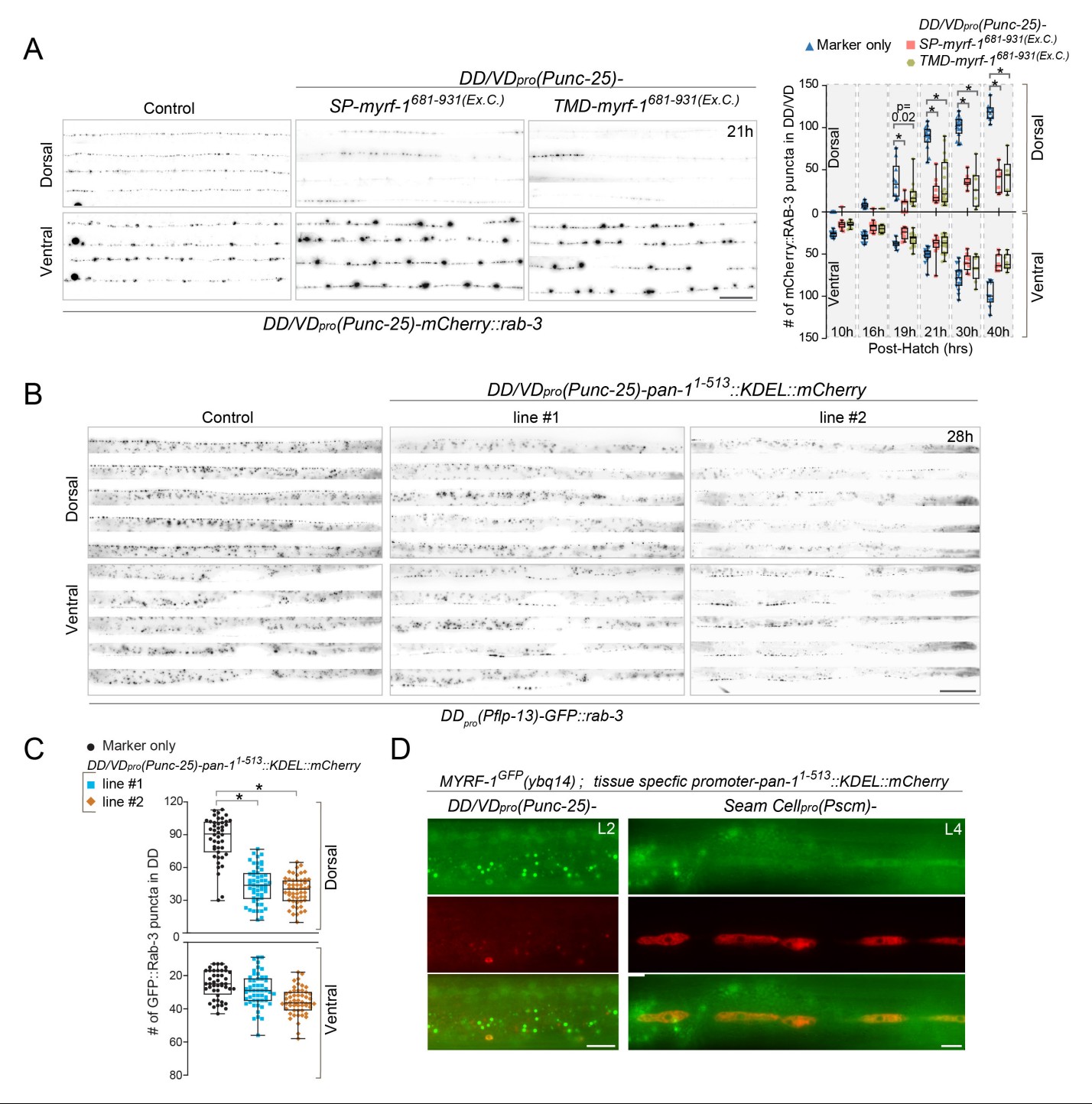

**Figure 11.** MYRF-PAN-1 interaction is important for synaptic remodeling. (**A**) Synaptic remodeling in animals with extracellular domain of MYRF overexpressed. Expressed in transgenes were *unc-25_pro_-SP-MYRF-1^Ex.C^.* (*ybqEx752*) and *unc-25_pro_-TMD-MYRF-1^Ex.C^.* (*ybqEx754*). SP-, 1-40aa of MIG-17. TMD-, 1-51aa of NEP-2. Ex.C., <u>extra</u>cellular region (681-931aa) of MYRF-1. Synaptic remodeling was labeled by *unc-25_pro_-mCherry::rab-3(juIs236)*. Scale bar, 20 μm. The graph shows the number of synapses at various post-hatch hours. Shown as Mean ± SEM (t test. *, p<0.001.). (**B**) Synaptic remodeling in animals with ER-retained PAN-1 overexpressed. Expressed in transgenes were *unc-25_pro_-pan-11^1-513^::KDEL::mCherry, line#1 ybqEx815, line#2 ybqEx817*. Synaptic remodeling was labeled by *flp-13_pro_-GFP::rab-3(ybqIs47)*. Images of five ventral and dorsal cords are vertically tiled. Scale bar, 20 μm. (**C**) Quantification of the number of synapses in (**B**), shown as Mean ± SEM (t test. *, p<0.0001.). (**D**) MYRF-1^GFP^ was trapped in cytoplasm by overexpression of ER-retained PAN-1. *unc-25_pro_-pan-11^1-513^::KDEL::mCherry(ybqEx807), SCM_pro_-pan-11^1-513^::KDEL::mCherry(ybqEx820)* was overexpressed in *myrf-1^GFP^(ybq14)* background. The MYRF-1^GFP^ signals were greatly increased in mCherry positive cells and localized in cytoplasm,
*Figure 11 continued on next page*

*Figure 11 continued*

while MYRF-1$^{GFP}$ was normally localized on cell membrane and nucleus at L2 (left panel), or largely down-regulated at L4 (right panel). The images were taken using wide-field microscope. Scale bar, 10 μm.

## Materials and methods

### Experimental model and subject details

#### Animals

Wild-type *C. elegans* were Bristol N2 strain. Strains were cultured on NGM plates using standard procedures (*Brenner, 1974*) at 20–23°C. Animals were cultured at 20°C for assays requiring specific developmental stages. Animals analyzed in this paper were hermaphrodite.

#### Cell lines

Human embryonic kidney cells 293 (HEK293T) were obtained from cell bank of CAS (Shanghai). The cells were verified by STR profiling and tested to be free of mycoplasma contamination by stand PCR methods.

### Method details

#### Naming of the alleles

All alleles generated in Y.B.Q. lab are designated as 'ybq' alleles, and all strains, as 'BLW' strains. 'Ex' denotes transgene alleles of exchromosomal array. 'Is' denotes integrated transgene. 'syb' alleles (in 'PHX' strains) are generated by genomic editing using CRISPR-Cas9 technique. 'syb' alleles were designed by Y.B.Q. and produced by SunyBiotech (Fuzhou, China).

#### Myrf alleles by CRISPR-Cas9 editing

For the following described alleles generated by CRISPR-Cas9 editing, Cas9 and gRNA were expressed from plasmids. The positive clones were identified using PCR screening to test singled F1 resulted from microinjection.

myrf-1$^{S483A\ K488A\ GFP}$(syb1487) allele was generated in the background of strain BLW889, GFP::myrf-1::3xFlag(ybq14). The resulted allele changes Ser 483 to Ala; Lys 488 to Ala. gRNA targets are:

> Sg1:CCGATATTCGACTCAAGGAAGCA;
> Sg2:TCCATCCGATATTCGACTCAAGG
> ybq14(background): . . .ATTTATATGAGTGGCCGAATAATCAATCCATCCGATATTCGACT-
> CAAGGAAGCAATTACTGAACGGGAAACTGCTGAA
> syb1487: . . .ATTTATATGAGTGGCCGAATAATCAATCCAGCTGATATCAGACTTGCAGAAG-
> CAATTACTGAACGGGAAACTGCTGAA

myrf-1$^{1-700\ GFP}$(syb1313) allele was generated in the background of strain BLW889, GFP::myrf-1::3xFlag(ybq14). The resulted allele removes G701-T927 of MYRF-1 coding sequence. gRNA targets are:

> Sg1:CCACTAAAGGAGAACTCGCCAAT;
> Sg2:CCTATCATCTTCAGACTACAAGG
> ybq14(background): . . .CATTTCGAAACGAACACACCATCCACTAAA-(951 bp)- CTATCATC
> TTCAGACTACAAGGACCACGAC
> syb1313: . . .CATTTCGAAACGAACACACCATCCACTAAA-(delete 951 bp)-CTATCATCTTCA-
> GACTACAAGGACCACGAC

myrf-1$^{1-790\ GFP}$(syb1333) allele was generated in the background of strain BLW889, GFP::myrf-1::3xFlag(ybq14). The resulted allele removes I791-C926 of MYRF-1 coding sequence. gRNA targets are:

> Sg1:CCTGCACTGAATGTGACACTTGA;
> Sg2:CCACGACGGTGACTACAAGGACC
> ybq14(background): . . .GAGCACATGGCTTTTGAAACTGGAGTTGAG-(551 bp)-ACCCTATCA
> TCTTCAGACTACAAGGACCAC

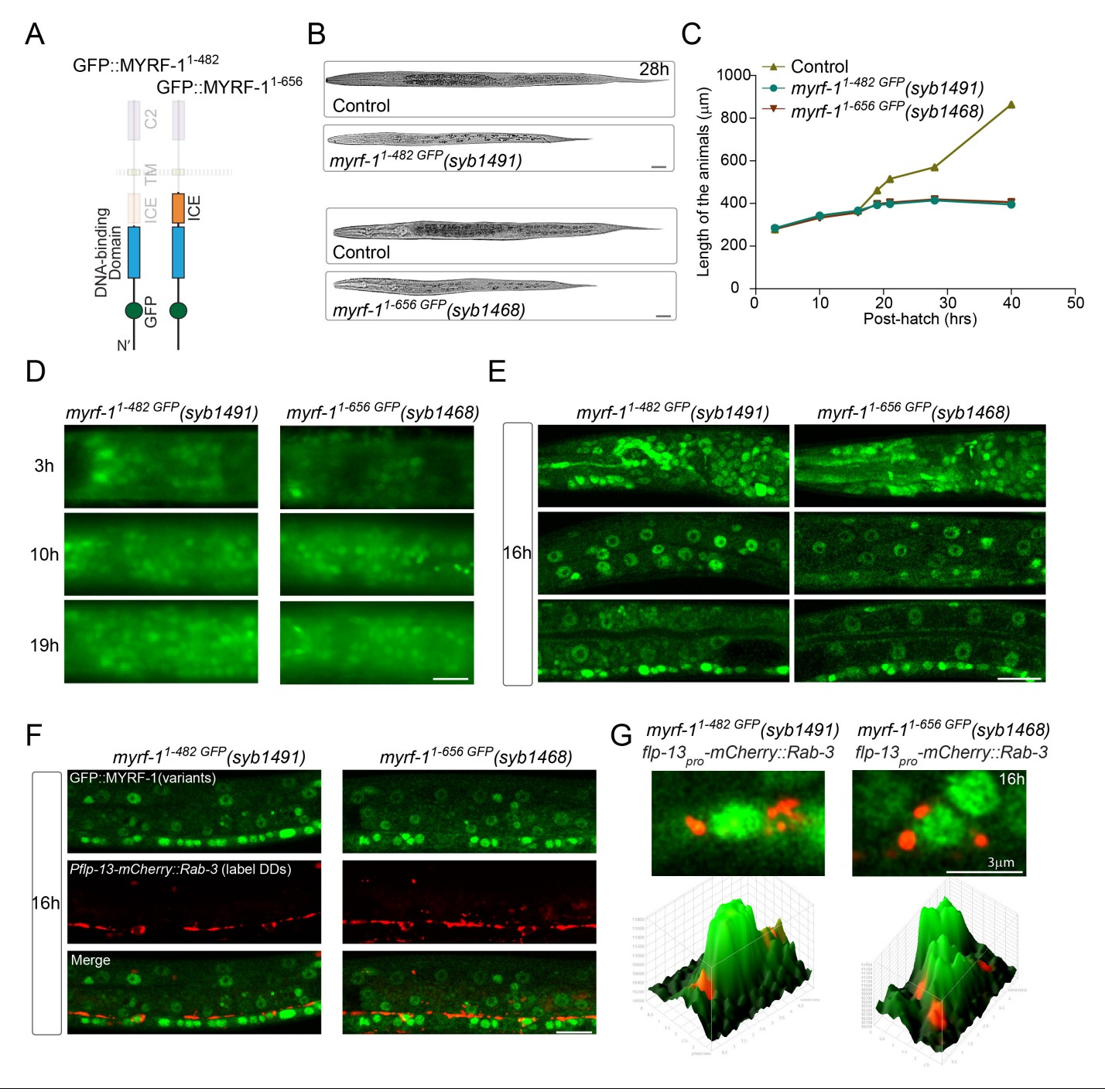

**Figure 12.** N-terminal MYRF alone is insufficient for MYRF's function. (**A**) Illustration of truncated MYRF-1 proteins encoded by *myrf-1*$^{1-482\ GFP}$*(syb1491)* and *myrf-1*$^{1-656\ GFP}$*(syb1468)*, with in-frame deletion of the whole (starting from cleavage point) and the part (starting from the end of ICE region) of the C-terminal region of MYRF-1, respectively. (**B**) DIC images of *myrf-1*$^{1-482\ GFP}$*(syb1491)* and control *myrf-1(syb1491)/mIn1* animals (top); *myrf-1*$^{1-656\ GFP}$*(syb1468)* and *myrf-1(syb1468)/mIn1* animals (bottom). At 28 post-hatch hours, the mutants were alive but shorter because they arrested at earlier stage. Scale bar, 20 µm. (**C**) Body length of *myrf-1*$^{1-482\ GFP}$*(syb1491)*,*myrf-1*$^{1-656\ GFP}$*(syb1468)*, and control animals *myrf-1(syb1491)/mIn1* was measured and shown as Mean, n = 15. (**D**) GFP signals in *myrf-1*$^{1-482\ GFP}$*(syb1491)* and *myrf-1*$^{1-656\ GFP}$*(syb1468)* mutants at 3, 10, 16 post-hatch hours. GFP signals were primarily observed in the nuclei (more than 20 animals were imaged and analyzed for each developmental stage). Images show head regions, and were acquired by wide-filed camera. Scale bar, 10 µm. (**E**) Single optical-plane images of MYRF-1$^{1-482\ GFP}$(*syb1491*) and MYRF-1$^{1-656\ GFP}$(*syb1468*) at 16 post-hatch hours acquired by Airyscan confocal microscopy. All sections are sagittal. Top row, head region. Middle row, lateral (left-right) section of the middle (head-tail) segment. Bottom row, medial (left-right) section of the middle (head-tail) segment with a focus on ventral nerve cord. Scale bar, 10 µm. (**F**) Colabeling MYRF-1$^{1-700\ GFP}$(*syb1313*) and MYRF-1$^{1-790\ GFP}$(*syb1333*) with DD neuron marker *Pflp-13-mCherry::rab-3(ybqIs1)* at 16 post-hatch

*Figure 12 continued on next page*

*Figure 12 continued*

hours. The animals carry *glo-1(zu391)* to reduce auto-fluorescence. Shown are single optical slices by Airyscan confocal microscopy. RAB-3 is localized in cytoplasm. White arrow, soma of DDs. Scale bar, 10 μm. (**G**) Shown are the magnified image of DD soma from the colabeling experiment (**F**), and 3-D intensity plots of green and red signals in the upper image.

The online version of this article includes the following figure supplement(s) for figure 12:

**Figure supplement 1.** Comparing the signal intensity in MYRF-1[1-482 GFP](*syb1491*) and MYRF-1[1-656 GFP](*syb1468*).

**Figure supplement 2.** Single-copy-transgene of MYRF-1[1-482] (N-MYRF-1) in DDs does not rescue synaptic rewiring defects.

syb1333: ...GAGCACATGGCTTTTGAAACTGGAGTTGAG-(delete 551 bp) <u>CCTATCATC TTCAGACTACAAGGACCAC</u>

myrf-1[1-482 GFP](*syb1491*) allele was generated in the background of strain BLW889, GFP::myrf-1::3xFlag(*ybq14*). The resulted allele removes Ser483-Ser931 and 3xFlag. gRNA targets are:

Sg1:CCGATATTCGACTCAAGGAAGCA;
Sg2:CCACGACGGTGACTACAAGGACC
ybq14(background): ...ATTTATATGAGTGGCCGAATAATCAATCCA-(1683 bp)-TAA
gctaatgcttcatatatcatttttgtt...
syb1491: ...ATTTATATGAGTGGCCGAATAATCAATCCA-(delete 1683 bp)-TAA
gctaatgcttcatatatcatttttgtt...

myrf-1[1-656 GFP](*syb1468*) allele was generated in the background of strain BLW889, GFP::myrf-1::3xFlag(*ybq14*). The resulted allele removes Gly657-Ser931 and 3xFlag. gRNA targets are:

Sg1:CCGACTTAGTCAAGGAACAGTTG;
Sg2:CCACGACGGTGACTACAAGGACC
ybq14(background): ...GCACAATCCTGTGGAAGCCGACTTAGTCAA-(1161 bp)-TAA
gctaatgcttcatatatcatttttgtt...
syb1468: ...GCACAATCCTGTGGAAGCCGACTTAGTCAA- (delete 1161 bp)- TAA
gctaatgcttcatatatcatttttgtt...

myrf-2[1-728 GFP](*syb1454*) allele was generated in the background of strain BLW1111, myrf-2 (*ybq46*)[GFP]. The resulted allele removes Leu729-Val949 of MYRF-2 coding sequence. gRNA targets are:

Sg1:CGTCGTTATTCCATTGGgtaagg;
Sg2:CCGCGCGTGCAATAGAACAAACT
ybq46(background): ...AAGGAAGGTCCAGGAAAC<u>GTC</u>GTTATTCCA-(1244 bp)-TAA
catcagagataatcacacaatagatac...
syb1454: ...AAGGAAGGTCCAGGAAAC<u>GTG</u>GTTATTCCA-(delete 1244 bp)-TAA
catcagagataatcacacaatagatac...

## pan-1[GFP] knock-in allele by CRISPR-Cas9 editing

pan-1GFP(*ybq47*) was generated in the background of wild-type (N2). GFP is inserted before stop codon (TAA) of the PAN-1 coding sequence. gRNA target:

sgRNA#2, GGAATTTTCAGCACGGTCAAAGG.
WT: ...AATACT<u>GGG</u>CCGAAGAAGACTGTACGATTCCAGAATTTT-(insertion point)-TAA
agaagaaaaatttcataacctcaaatca
ybq47: ...AATACT<u>GGA</u>CGAAGAAGACTGTACGATTCCAGAATTTT-GFP-TAA
agaagaaaaatttcataacctcaaatca.

## pan-1[LoxP] allele by CRISPR-Cas9 editing

pan-1[LoxP](*syb1217*) allele was generated in the background of wild-type (N2). Two LoxP sites are inserted into flanking introns of the third exon of *pan-1b* gene. gRNA target sites: Sg1:CCAGTA TGCATCTGCGCCGACAA; Sg2:GTATGCATCTGCGCCGACAACGG; Sg3:CAAGTTAAAC TCGAACGATGTGG; Sg4:CCAAATTCCTTCCAATCCCTTGG; Cas9 and gRNA were expressed from plasmids. The repair template is plasmid-based. The positive clones were identified using PCR screening to test singled F1 resulted from microinjection.

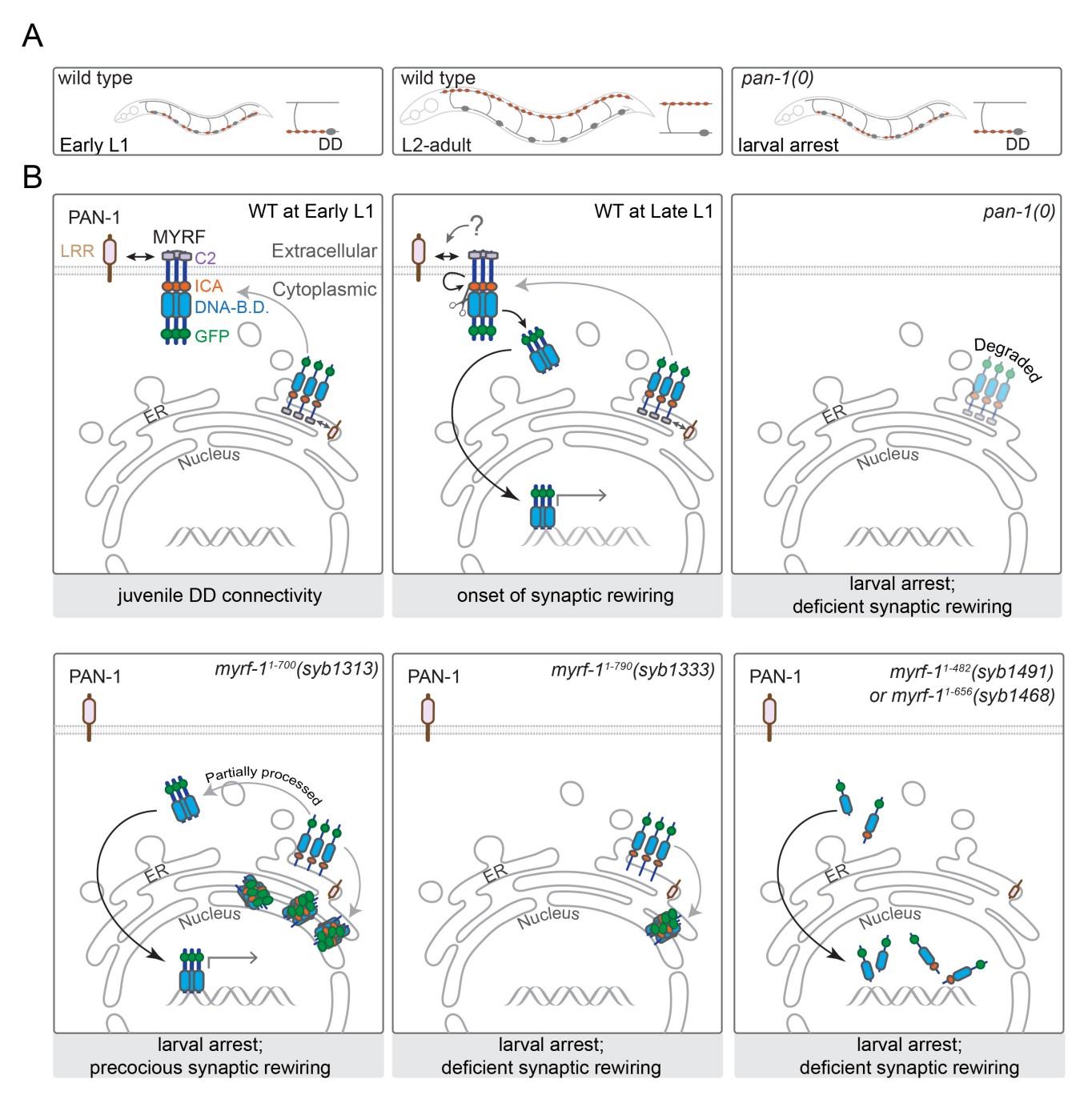

**Figure 13.** A model for how PAN-1 and MYRF interact to regulate synaptic rewiring. (**A**) Illustration of synaptic rewiring in wild type and *pan-1(0)* mutants. Synapses in DDs are indicated by brown-color dots. (**B**) PAN-1 and MYRF interaction via extracellular domains is required for cell-membrane localization of MYRF. The release of N-MYRF occurs at late L1, coinciding the timing when MYRF is critically required for driving L1-L2 larval transition and synaptic remodeling in DD neurons. In the absence of PAN-1, MYRF is degraded, which leads to larval arrest and blocked DD synaptic remodeling. MYRF-1$^{1-700}$(*syb1313*) is partially processed in ER, which produces N-MYRF translocating into nucleus, even though a large amount of MYRF-1$^{1-700}$(*syb1313*) accumulates in ER. MYRF-1$^{1-790}$(*syb1333*) is not cleaved; they reside in ER and are mostly degraded. MYRF-1$^{1-482}$(*syb1491*) and MYRF-1$^{1-656}$(*syb1468*), neither of which contains TMD, are transported into nucleus but fail to regulate gene expression.

WT: . . .
atacctgtaacttttcagCCAGTATGCATC<u>TGC</u>GCCGACAACGGAATTTTCAGCACGGTCAAAGGA
TTTACTATAGAgtatgttgaattttact-(insertion      point)-cgactataaaacttcttgaaactgaaattttcttttcagA
TGCGAATCCGCTTCAATTGCATCTGTATCTGAAAATTTGGCTTCACTGAATGGCACAGAA
TTGGGACGTCTCACAATAAGAGATTCTACAGTTAATGTTCTTCCACAGGATCTTTTTGAGAA
T**g**taagttgaattacaataatgccaatcaggacagaaacgtatatc-(insertion          point)-actagttaactttttct-
catcgcggcgagacccacttaaatcaatccaattttcagGTCTTCGCTAAACAAGTT<u>AAA</u>CTCGAACGATG
TGGTCTTTCAACTCTTCAACCA<u>AAT</u>TCCTTCCAATCCCTTGGAGGATCAGCTGAA
syb1217: . . .
atacctgtaacttttcagCCAGTATGCATC<u>TGT</u>GCCGACAACGGAATTTTCAGCACGGTCAAAGGA
TTTACTATAGAgtatgttgaattttact-(ATAACTTCGTATAGCATACATTATACGAAGTTAT)-cgacta-
taaaacttcttgaaactgaaattttcttttcagATGCGAATCCGCTTCAATTGCATCTGTATCTGAAAA
TTTGGCTTCACTGAATGGCACAGAATTGGGACGTCTCACAATAAGAGATTCTACAGTTAATG
TTCTTCCACAGGATCTTTTTGAGAAT**g**taagttgaattacaataatgccaatcaggacagaaacgtatatc-(A
TAACTTCGTATAGCATACATTATACGAAGTTAT)-actagttaactttttctcatcgcggcgagacccacttaaat-
caatccaattttcagGTCTTCGCTAAACAAGTT<u>AAG</u>CTCGAACGATGTGGTCTTTCAACTC
TTCAACCA<u>AAC</u>TCCTTCCAATCCCTTGGAGGATCAGCTGAA.

## Generation of transgene alleles
### DD/VD – pan-1 isoforms rescue
unc-25$_{pro}$-pan-1b(ybqIs130): The vector pQA1237 was injected into N2 at 20 ng/μl in N2. The resulted Ex array was integrated to generate ybqIs130. unc-25$_{pro}$-pan-1a(ybqEx624): The vector pQA1239 was injected into BLW1167 [unc-25$_{pro}$-mCherry::rab-3(juIs236)/mT1; pan-1(gk142)/mT1] at 20 ng/μl. unc-25$_{pro}$-pan-1c(ybqEx626): The vector pQA1240 was injected into BLW1167 at 20 ng/μl.

### DD:
flp-13$_{pro}$-pan-1b(ybqEx821): The vector pQA1284 was injected into BLW1056 [unc-25$_{pro}$-mCherry::rab-3(juIs236)/mT1; pan-1(gk142)/mT1], at 5 ng/μl.

### Epidermis:
dpy-7$_{pro}$-pan-1b(ybqEx741): The vector pQA1298 was injected into BLW1056 at 10 ng/μl.

### Body wall muscle:
myo-3$_{pro}$-pan-1b(ybqEx737): The vector pQA1297 was BLW1056 at 10 ng/μl.

### Intestine
vha-6$_{pro}$-pan-1(ybqEx739): The vector pQA1583 was injected into BLW1056 at 10 ng/μl.

### pan-1 promoter reporter
pan-1$_{pro}$-NLS-tagRFP(ybqIs138): The vector pQA1311 was injected into N2 at 20 ng/μl, and the Ex array transgene was integrated to generate ybqIs138.

### Seam cell
seam-cell$_{pro}$-pan-1b::mCherry(ybq Ex778).The vector pQA1534 was injected into BLW1465 [myrf-1$^{GFP}$(yqb14); pan-1(gk142)/mT1] at 20 ng/μl.

### DD/VD$_{pro}$-nCre
unc-25$_{pro}$-nCre(tmIs1073) was sent by National Bioresource Project for the Experimental Animal 'Nematode *C. elegans*' (*Kage-Nakadai et al., 2014*).

### pan-1 deletion mutant rescue:
The following vectors were injected into BLW1056 at 50 ng/μl. pQA1322 Punc-25-pan-1::GFP-Δ E130-S442 (almost all LRR region) for *unc-25$_{pro}$-pan-1ΔLRR(ybqEx642)*; pQA1323 Punc-25-pan-1:: GFP-ΔR2-S18 (SP) for *unc-25$_{pro}$-pan-1ΔSP(ybqEx643)*; pQA1324 Punc-25-pan-1::GFP-ΔG513-F538 (TM) for *unc-25$_{pro}$-pan-1ΔTM (ybqEx644)*; pQA1337 Punc-25-pan-1::GFP-ΔK539-F594 (Cyto) for *unc-25$_{pro}$-pan-1ΔCyto(ybqEx645)*.

Overexpress MYRF-1681$^{681-931}$:

The following vectors were injected into CZ8656 [Punc-25-mCherry::Rab-3(juIs236)] at 50 ng/µl. pQA1539 Punc-25-mig-17(1-40)::myrf-1 (681–931) for *unc-25$_{pro}$-SP-MYRF-1$^{Ex.C}$.* (*ybqEx752*); pQA1540 Punc-25-nep-2 (1–51)::myrf-1 (681–931) for *unc-25$_{pro}$-TMD-MYRF-1$^{Ex.C}$*. (*ybqEx754*).

Overexpress ER-retained pan-1:

The vector pQA1654 was injected into BLW1419 [Pflp-13-GFP::rab-3(ybqIs47)] at 20 ng/µl to make Punc-25-pan-1(1-G513)::mCherry::KDEL(ybqEx815); Punc-25-pan-1(1-G513)::mCherry::KDEL (ybqEx817). The vector pQA1654 was injected into CZ8761 [Punc-25-mCherry::Rab-2(juIs236); Pacr-2-GFP(juIs14)] to make Punc-25-pan-1(1-G513)::mCherry::KDEL(ybqEx807), and ybqEx807 was crossed into BLW889 to make BLW1890. The vector pQA1652 was injected into BLW889 to make Pscm-Pan-1(1-G513)::mCherry::KDEL(ybqEx820).

## Knocking down ubiquitination-mediated proteasome components by RNAi

HT115(DE3) strains carrying empty vector L4440, ubq-2, or rpt-2 plasmids (from *C. elegans* of RNAi library generated by M. Vidal et al.) were cultured in LB with 100 µg/ml of Ampicillin for 16–18 hr at 37°C. About 50 µl of bacteria liquid culture was seeded onto a NGM plate with 100 µg/ml of Ampicillin and 1 mM of IPTG (Takara) for induction of dsRNA expression. Stage-synchronized early L1 animals were cultured on RNAi plates at 20°C for 24 hr. The animals were then examined and imaged under OLYMPUS BX63 microscope.

## RT-qPCR

About 600 synchronized pan-1(0) mutants (BLW1107: unc-25$_{pro}$-mCherry::rab-3(juIs236); pan-1 (gk142); hlh-8$_{pro}$-GFP(ayIs6)) of L2 that were hand-picked, together with 50 µl of the stage-matched control animals (BLW817: juIs236; ayIs6) were collected and frozen in the RNAiso plus reagents (Takara). Total RNAs were extracted and the cDNAs were synthesized using the reverse transcription system (Takara). Quantitative PCR reactions were performed in triplicates on a Quantstudio 7 PCR machine using the SYBR Green dye (Vazyme). Relative gene expression levels were calculated using the 2−ΔΔCt method and plotted as median with range48M. RT-qPCR experiments were performed at least three times using RNA samples derived from animal tissues collected independently. The data were analyzed in GraphPad Prism. qPCR primers include: myrf-1, gacgatggctcaatggtgtc(F), ttgggtccgaatgatgtgat(R), 214 bp; myrf-2, ccgatgacatcccaaatcgcaaac(F), gaagagttgggtcgtctccaag(R), 204 bp, pan-1, cctcccaagtccgtactccattc(F), ccattccttccaatgcgcgag(R), 174 bp; pmp-3, tggccggatgatggtgtcgc(F), acgaacaatgccaaaggccagc(R), 190 bp.

## Microscopic analysis and quantification

Live animals were anesthetized using 0.5% 1-Phenoxy-2-propanol in M9 and mounted on 3% Agar gel pad. The animals were examined under x60 oil objective on OLYMPUS BX63 microscope. The wide-field DIC or fluorescence images (single plane images or Z-focal stacks) were taken by a sCMOS camera (PrimΣ Photometrics camera (model 2)) mounted on OLYMPUS BX63, which is driven by CellSens Dimension software (Olympus). Images of live animals were also acquired on Zeiss LSM880 with Airyscan. The thickness of the optical slices was typically 0.8 µm.

For quantification of patterns of GFP::MYRF, images of stage-synchronized animals were acquired using wide-field microscopy described as above. The same parameters were used, including the power of excitation light, identical objective, exposure duration, and screen display setting. The images are examined and the patterns of GFP::MYRF were categorized based on how consistently the signals are observed at particular subcellular localization throughout the animal body. 'Weak or unclear signal' means that no clear signals are detected, or there are some weak signals but they are inconsistent throughout the animal body. Three independent rounds of culture, imaging, and scoring were carried out, and the data were pooled and presented in the percentage column graphs.

For generation of the 3-D plot of MYRF-1$^{GFP}$ (and variants) and RFP-Rab-3 colabeling, each image acquired from Zeiss Airyscan was opened in ImageJ in color mode, a rectangular ROI was selected to contain a complete DD soma, a plug-in program '3-D surface plot' in Analyze menu was run with the following parameters, Filled; Grid Size 128; Smoothing 1.5; Perspective 0; Lighting 0.2 to generate the plot image.

For generation of the line plot of MYRF-1(syb1313)$^{1-700\ GFP}$ and MYRF-1(syb1333)$^{1-790\ GFP}$, each image stack acquired from Zeiss Airyscan was opened in ImageJ and a single slice with focused seam cell was selected for further analysis. A 20 pixel (~1 µm)-thick, 12-µm-long ROI line was drawn to cross a seam cell in head-to-tail direction. The middle point of the ROI line was positioned at the center of the nucleus. A ROI for analyzing single ventral cord neuron was similarly drawn except for that the line was 5µm-long. A program 'plot profile' located in Analyze menu was run and plot data was recorded. The final graph was generated in GraphPad Prism 6. Each plot line represented the intensity distribution across a single cell from an independent animal.

Quantification of patterns of PAN-1$^{GFP}$ were carried out similarly to GFP::MYRF. Whether PAN-1$^{GFP}$ are consistently observed on cell membrane throughout the animal body was examined and quantified. For quantification of intensity of PAN-1$^{GFP}$ in myrf-1(ju1121) mutants, images of PAN-1$^{GFP}$ in myrf-1 mutants and controls were acquired by wide-field microcopy similarly to GFP::MYRF, and then analyzed in ImageJ. A ROI of 5 pixel (~0.8 µm)-thick line was drawn along the PAN-1$^{GFP}$ signals at the cell membrane of seam cells. For images acquired from Zeiss Airyscan, each image was opened in ImageJ, and a ROI of 5 pixel (~250 nm)-thick line was drawn along the PAN-1$^{GFP}$ signals at the cell membrane of seam cells. The mean intensity of the ROI was measured and recorded. At least two seam cells were measured in each animal. A single data point represented the mean intensity of PAN-1$^{GFP}$ signals for each animal.

## Quantification of synaptic remodeling

Bleach-synchronized arrest young L1 were cultured at 20°C to appropriate post-hatch hours. Animals were anesthetized, and imaged on OLYMPUS BX63 microscope using wide field sCMOS camera. The z-stack was maximally projected to produce a single image in ImageJ, or single section was analyzed when Z-maximum-projection did not produce clear image (e.g. the animal shifted). The complete ventral and dorsal cords were selected using Segmented Line tool to generate a line ROI of 30 pixel-wide (~5 µm). Run Plot Profile under 'Analyze' menu. Run Find Peaks under BAR-Data Analyze, set values in 'Find Local Maxima/Minima' window so that the program detects all visible synapses while excluding minor peaks, and execute the counting. Signals of each cord were also examined by eyes to confirm there was no major discrepancy between human evaluation and computer counting. The values and statistics are then processed in GraphPad Prism. The mean number of puncta for individual group was presented in the scatter dot plot graphs.

## Quantification of animal length

For the analysis of the growth of the animals, DIC images of animals were acquired on OLYMPUS BX63 microscope using x10 objective. The lengths of animals in the images were measured using polyline tool in the OLYMPUS imaging software. The data were further analyzed in GraphPad Prism.

## Constructs used in transfection experiments

All constructs used in transient expression in HEK293T cells were built in pcDNA3.1 with CMV promoter. Full-length myrf-1 was mammalian codon optimized and encodes C. elegans myrf-1 protein. The DNA V5::myrf-1::3xFlag (in pQA1228) was synthesized by Genscript. 6xHis::3xFlag::myrf-1 (1–482) (in pQA1250) and myrf-1 (483–931)::3xFlag (in pQA1296) were made by mutagenesis from pQA1228. myrf-1 (680–931)::3xFlag (in pQA1525) and myrf-1 (586–931): :3xFlag (in pQA1581) were made by mutagenesis from pQA1296.

Full-length pan-1b was cloned by PCR from C. elegans cDNA and inserted into pcDNA3.1 (pan-1::HA::His in pQA1269; with addition of Kozak sequence in pQA1524). pan-1 (19–594) (ΔSP) (in pQA1527), pan-1 (19–512; 539–594) (ΔSP, TMD) (in pQA1580), pan-1 (1–512) (ΔTMD, Cyto) (in pQA1528), and pan-1 (1–78; 444–594) (ΔLRR) (in pQA1535) were made by mutagenesis from pQA1524 and they all have HA-His tag at C'.

## Transient transfection in cell line and co-immunoprecipitation

HEK293T cells were cultured in DMEM supplied with 10% Foundation FBS (Gemini) and 100 units/ml penicillin/streptomycin (HyClone) antibiotics. in six well-plates. For transient transfection, adherent cells of 0.5–2 × 10$^5$ per well were seeded in 500 µl antibiotic-free medium on the day before transfection. The cells were let grown to 90–95% confluence by the time of transfection. Discard the old

medium before transfection. Prepare transfection mix following the recipe: For each well, dilute plasmid DNA (2.5 μg) in 150 μl DMEM serum-free medium and mix gently. The following was an example of set up: (1) Plasmid A(1.25 μg)+Control plasmid(1.25 μg); (2) Plasmid B(1.25 μg)+Control plasmid(1.25 μg); (3) Plasmid A (1.25 μg)+Plasmid B(1.25 μg). Gently shake Lipofectamine-2000 (Thermo Fisher) well before use. Take an appropriate amount of Lipofectamine-2000 (6 μl) and dilute it in 150 μl DMEM medium. Incubate at room temperature for 5 min. Mix the DNA and Lipofectamine-2000 to make the total volume 300 μl, gently mix and leave at room temperature for 20 min. Add 300 μl of transfection solution to each well of cells and shake gently. The cells were returned to 37℃, and the medium was changed after 4–6 hr of transfection.

The transfected cells were incubated at 37℃ for 24 hr. The cells were lysed in 1 mL of cell lysis buffer (50 mM Tris PH7.4, 150 mM NaCl, 1% NP40, 0.25% Sodium taurodeoxycholate hydrate, 1 mM EDTA, protease inhibitor cocktail). The lysate was incubated on ice for 30 min and centrifuged at 14,000Xg for 30 min at 4℃. The supernatant was collected and incubated with addition of primary antibodies, including anti-HA (ab9110, Abcam, 1:1000) and rabbit IgG (2729, Cell Signaling) controls at the same molar concentration as primary antibodies, respectively, at 4℃ for overnight. The next day, Dynabeads protein A (10002D, invitrogen) were added to the reactions and incubated at 4℃ for 1 hr to precipitate the IgGs. Beads were washed and treated in SDS sample loading buffer. The samples were denatured at 95℃ for 10 min and analyzed by SDS-PAGE and western blot. The antibodies used in western blot included: mouse anti-V5 (ab27671, Abcam, 1:5000); mouse anti-HA (ab18181, Abcam, 1:5000); Rabbit Anti-Actin (ab179467, 1:5000); Mouse Anti-Mouse IgG Polyclonal Antibody, HRP Conjugated (Genscript,1:20000), Mouse Anti-Rabbit IgG Polyclonal Antibody, HRP Conjugated (Genscript,1:20000).

## Immunoprecipitation followed by mass spectrometry analysis

Worms expressing the GFP-tagged proteins were cultured to the indicated stage, collected, and washed three times with M9 buffer. The volumes of the worm pellets obtained after centrifugation and sedimentation on ice was recorded before the worm pellets were frozen in liquid nitrogen. The frozen worm pellets were cryo-milled using Mixer Mill MM 400 (Retsch) at 30 Hz for 60 s each time for a total of four times. The resulting powder for each sample was transferred to a new tube, mixed with an equal volume of $2 \times$ pre cool lysis buffer containing 50 mM Tris-HCl pH 8.0, 300 mM NaCl, 1% Triton-X100, 10% glycerol, 1 mM PMSF, $2 \times$ Roche cOmplete EDTA-free Protease Inhibitor Cocktail, 10 mM NaF, and let rotate at 4℃ for 30 min. The lysate was then centrifuged at 13,000 rpm for 30 min at 4℃ and the supernatant was incubated with homemade anti-GFP beads for one hour at 4℃ before the beads were washed three times with cold lysis buffer. The anti-GFP immunoprecipitates were eluted with chilled elution buffer (100 mM glycine-HCl pH 2.5) and precipitated with 25% TCA. After cold acetone wash and drying in air, the immunoprecipitated proteins were dissolved in 100 mM Tris, pH 8.5, 8 M urea, reduced with 5 mM TCEP, alkylated with 10 mM iodoacetamide, and digested with trypsin overnight at 37℃.

LC-MS/MS analysis of the resulting peptides was conducted on a Q Exactive mass spectrometer (ThermoFisher Scientific) interfaced with an Easy-nLC1000 liquid chromatography system (ThermoFisher Scientific). Peptides were loaded on a pre-column (75 μm ID, 6 cm long, packed with ODS-AQ 120 Å −10 μm beads from YMC Co., Ltd.) and separated on an analytical column (75 μm ID, 13 cm long, packed with Luna C18 1.8 μm 100 Å resin from Welch Materials) using an 0–30% (v/v) acetonitrile gradient over 78 min at a flow rate of 200 nl/min, with 0.1% (v/v) formic acid added to the mobile phase throughout. The top 15 most intense precursor ions from each full scan (resolution 70,000) were isolated for HCD MS2 (resolution 17,500; NCE 27), with a dynamic exclusion time of 30 s. Precursors with unassigned charge states or charge states of 1+, 7+, or > 7+ were excluded. Each sample was analyzed in technical duplicates.

Peptides and proteins were identified from the MS data using the database searching engine pFind 3.1 (http://pfind.ict.ac.cn/) against the UniProt *C. elegans* protein database (WS235). The filtering criteria were: 1% FDR at both the peptide level and the protein level; precursor mass tolerance, 20 ppm; fragment mass tolerance, 20 ppm; the peptide length, 6–100 aa. To identify high-confidence candidate interacting proteins, The WD-Scores of proteins identified in each IP-MS sample were calculated against the identification results of 71 unrelated IP-MS samples, according to the formulas from the CompPASS tutorial (*Sowa et al., 2009*).

## Acknowledgements

Some strains were provided by the CGC, which is funded by NIH Office of Research Infrastructure Programs (P40 OD010440). This work was supported by the National Natural Science Foundation of China (NFSC) general programs #31571272 to YBQ; the lab start-up fund from ShanghaiTech University to YBQ; MQD and YHY are supported by the National Natural Science Foundation of China #32061143020 and an institutional funding of NIBS, Beijing/TIMBR, whose funding source include the municipal government of Beijing, the Ministry of Science and Technology of the People's Republic of China, and Tsinghua University. We thank the Molecular Imaging Core Facility (MICF) at the School of Life Science and Technology, ShanghaiTech University for providing technical support.

## Additional information

### Funding

| Funder | Grant reference number | Author |
| --- | --- | --- |
| National Natural Science Foundation of China | 31571272 | Yingchuan B Qi |
| ShanghaiTech University | | Yingchuan B Qi |
| Ministry of Science and Technology of the People's Republic of China | | Meng-Qiu Dong |
| National Natural Science Foundation of China | 32061143020 | Meng-Qiu Dong |
| Beijing Municipal Science and Technology Commission | | Meng-Qiu Dong |

The funders had no role in study design, data collection and interpretation, or the decision to submit the work for publication.

### Author contributions

Shi-Li Xia, Data curation, Formal analysis, Investigation, Analysis of pan-1 expression and cell-autonomy, Quantification of synaptic rewiring; Meng Li, Data curation, Formal analysis, Investigation, Analysis of new myrf-1 mutants' arrest, Initial analysis of MYRF and MYRF variants' localization; Bing Chen, Data curation, Formal analysis, Investigation, Co-IP analysis of MYRF-PAN-1 interaction; Chao Wang, Investigation, Initial analysis of synaptic rewiring in pan-1; Yong-Hong Yan, Investigation, Methodology, IP-MS analysis; Meng-Qiu Dong, Funding acquisition, Investigation, Methodology, IP-MS analysis; Yingchuan B Qi, Conceptualization, Data curation, Formal analysis, Supervision, Funding acquisition, Validation, Investigation, Visualization, Methodology, Writing - original draft, Project administration, Writing - review and editing, Analysis of synaptic rewiring in pan-1 and all new myrf-1 mutants, Analysis of PAN-1, MYRF and MYRF variants' subcellular localization, etc

### Author ORCIDs

Meng-Qiu Dong ⓘD http://orcid.org/0000-0002-6094-1182
Yingchuan B Qi ⓘD https://orcid.org/0000-0002-4267-4770

### Decision letter and Author response

Decision letter https://doi.org/10.7554/eLife.67628.sa1
Author response https://doi.org/10.7554/eLife.67628.sa2

## Additional files

### Supplementary files

- Supplementary file 1. Complete list for factors interacting with MYRF-1 in IP-MS analysis (in *Figure 1A*).

- Supplementary file 2. Complete list for factors interacting with MYRF-2 in IP-MS analysis (in *Figure 1A*).

- Supplementary file 3. Complete list for factors interacting with PAN-1 in IP-MS analysis (in *Figure 1A*).

- Supplementary file 4. p-values for comparison between all genotypes in *Figure 6*.

- Transparent reporting form

## Data availability

All data generated or analyzed during this study are included in the manuscript including supplementary files.

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

# Appendix 1

**Appendix 1—key resources table**

| Reagent type (species) or resource | Designation | Source or reference | Identifiers | Additional information |
|---|---|---|---|---|
| Strain, strain background (*C. elegans*) | Punc-25-mcherry::rab-3(juIs236) II | Yishi Jin lab | CZ8656 | |
| Strain, strain background (*C. elegans*) | Punc-25-mCherry::Rab-3(juIs236)/mT1 II, pan-1(gk142)/mT1 III | this paper | BLW1056 | |
| Strain, strain background (*C. elegans*) | +/mT1 II; pan-1(gk142)/mT1 III; Pflp-13-GFP::rab-3(ybqIs46) | this paper | BLW1115 | |
| Strain, strain background (*C. elegans*) | Punc-25-GFP(juIs76)/mT1 II, pan-1(gk142)/mT1 III | this paper | BLW1104 | |
| Strain, strain background (*C. elegans*) | Punc-25-mCherry::Rab-3(juIs236)/mT1 II, pan-1(gk142)/mT1 III; Punc-25-pan-1b::gfp (ybqIs131) | this paper | BLW1363 | |
| Strain, strain background (*C. elegans*) | Punc-25-mCherry::Rab-3(juIs236)/mT1 II, pan-1(gk142)/mT1 III; Punc-25-pan-1a (ybqEx624) | this paper | BLW1319 | |
| Strain, strain background (*C. elegans*) | Punc-25-mCherry::Rab-3(juIs236)/mT1 II, pan-1(gk142)/mT1 III; Punc-25-pan-1c (ybqEx626) | this paper | BLW1321 | |
| Strain, strain background (*C. elegans*) | Punc-25-mCherry::Rab-3(juIs236)/mT1 II, pan-1(gk142)/mT1 III | | BLW1167 | |
| Strain, strain background (*C. elegans*) | pan-1-loxp (syb1217) III; Punc-25-mCherry::Rab-3(juIs236) II | this paper | BLW1646 | |
| Strain, strain background (*C. elegans*) | pan-1-loxp (syb1217) III; Punc-25-mCherry::Rab-3(juIs236) III; Punc-25-Cre (tmIs1072) | this paper | BLW1645 | |
| Strain, strain background (*C. elegans*) | myrf-1(ju1121) Punc-25-mCherry::Rab-3(juIs236) II; Pmyrf-1-gfp::myrf-1::flag-myrf-1_3'UTR(ybqEx164) | *Meng et al., 2017* | BLW314 | |
| Strain, strain background (*C. elegans*) | myrf-2(ybq42) X; Pmyrf-2-gfp::myrf-2::HA(ybqIs128) | this paper | BLW1348 | |
| Strain, strain background (*C. elegans*) | Ppan-1-NLS::tagRFP(ybqIs138) | this paper | BLW1397 | |
| Strain, strain background (*C. elegans*) | Punc-25-mCherry::Rab-3(juIs236)/mT1 II, pan-1(gk142)/mT1 III; Punc-25-pan-1b (delta LRR) (ybqEx642) | this paper | BLW1401 | |
| Strain, strain background (*C. elegans*) | Punc-25-mCherry::Rab-3(juIs236)/mT1 II, pan-1(gk142)/mT1 III; Punc-25-pan-1b (delta SP) (ybqEx643) | this paper | BLW1402 | |
| Strain, strain background (*C. elegans*) | Punc-25-mCherry::Rab-3(juIs236)/mT1 II, pan-1(gk142)/mT1 III; Punc-25-pan-1b (delta TM) (ybqEx644) | this paper | BLW1403 | |
| Strain, strain background (*C. elegans*) | Punc-25-mCherry::Rab-3(juIs236)/mT1 II, pan-1(gk142)/mT1 III; Punc-25-pan-1b (delta Cyto) (ybqEx645) | this paper | BLW1404 | |

*Continued on next page*

*Appendix 1—key resources table continued*

| Reagent type (species) or resource | Designation | Source or reference | Identifiers | Additional information |
|---|---|---|---|---|
| Strain, strain background (*C. elegans*) | Punc-25-mCherry::Rab-3(juIs236) II; Punc-25-mig-10(1-40)::myrf-1 (681–931)::Flag (ybqEx752) | this paper | BLW1674 | |
| Strain, strain background (*C. elegans*) | Punc-25-mCherry::Rab-3(juIs236) II; Punc-25-nep-2 (1–51)::myrf-1 (681–931)::Flag (ybqEx754) | this paper | BLW1676 | |
| Strain, strain background (*C. elegans*) | gfp::myrf-1 (1–700)::3xflag(syb1313)/mIn1 II | this paper | PHX1313 | |
| Strain, strain background (*C. elegans*) | gfp::myrf-1 (1–790)::3xflag(syb1333)/mIn1 II | this paper | PHX1333 | |
| Strain, strain background (*C. elegans*) | gfp::myrf-1 (1–482)::3xflag(syb1491)/mIn1 II | this paper | PHX1491 | |
| Strain, strain background (*C. elegans*) | gfp::myrf-1 (1–656)::3xflag(syb1468)/mIn1 II | this paper | PHX1468 | |
| Strain, strain background (*C. elegans*) | gfp::myrf-1(S483A,K488A)::3xflag(syb1313)/mIn1 II | this paper | PHX1487 | |
| Strain, strain background (*C. elegans*) | gfp::myrf-2 (1–728)::3xflag(syb1454) X | this paper | PHX1454 | |
| Strain, strain background (*C. elegans*) | gfp::myrf-1 (1–700)::3xflag(syb1313)/mIn1 II; Pflp-13-GFP::rab-3(ybqIs47) IV | this paper | BLW1570 | |
| Strain, strain background (*C. elegans*) | gfp::myrf-1 (1–790)::3xflag(syb1333)/mIn1 II; Pflp-13-GFP::rab-3(ybqIs47) IV | this paper | BLW1571 | |
| Strain, strain background (*C. elegans*) | gfp::myrf-1 (1–482)::3xflag(syb1491)/mIn1 II; Pflp-13-GFP::rab-3(ybqIs47) IV | this paper | BLW1617 | |
| Strain, strain background (*C. elegans*) | gfp::myrf-1 (1–656)::3xflag(syb1468)/mIn1 II; Pflp-13-GFP::rab-3(ybqIs47) IV | this paper | BLW1615 | |
| Strain, strain background (*C. elegans*) | gfp::myrf-1(S483A,K488A)::3xflag(syb1313)/mIn1 II; Pflp-13-GFP::rab-3(ybqIs47) IV | this paper | BLW1616 | |
| Strain, strain background (*C. elegans*) | gfp::myrf-1 (1–700)::3xflag(syb1313)/mIn1 II; myrf-2(ybq42) X; Pflp-13-GFP::rab-3 (ybqIs47) IV | this paper | BLW1756 | |
| Strain, strain background (*C. elegans*) | gfp::myrf-1 (1–790)::3xflag(syb1333)/mIn1 II; myrf-2(ybq42) X; Pflp-13-GFP::rab-3(ybqIs47) IV | this paper | BLW1779 | |
| Strain, strain background (*C. elegans*) | gfp::myrf-1 (1–482)::3xflag(syb1491)/mIn1 II; myrf-2(ybq42) X; Pflp-13-GFP::rab-3(ybqIs47) IV | this paper | BLW1760 | |
| Strain, strain background (*C. elegans*) | gfp::myrf-1 (1–656)::3xflag(syb1468)/mIn1 II; myrf-2(ybq42) X; Pflp-13-GFP::rab-3(ybqIs47) IV | this paper | BLW1758 | |
| Strain, strain background (*C. elegans*) | gfp::myrf-1(S483A,K488A)::3xflag(syb1313)/mIn1 II; myrf-2(ybq42) X; Pflp-13-GFP::rab-3 (ybqIs47) IV | this paper | BLW1759 | |

*Continued on next page*

*Appendix 1—key resources table continued*

| Reagent type (species) or resource | Designation | Source or reference | Identifiers | Additional information |
|---|---|---|---|---|
| Strain, strain background (*C. elegans*) | gfp::myrf-1(S483A,K488A)::3xflag (syb1313)/mln1 II; Pflp-13-mCherry::rab-3(ybqIs1) | this paper | BLW1791 | |
| Strain, strain background (*C. elegans*) | gfp::myrf-1 (1–700)::3xflag(syb1313)/ mln1 II; glo-1(zu391) X; Pflp-13-mCherry::rab-3(ybqIs1) | this paper | BLW1833 | |
| Strain, strain background (*C. elegans*) | gfp::myrf-1 (1–790)::3xflag(syb1333) /mln1 II; glo-1(zu391) X; Pflp-13-mCherry::rab-3(ybqIs1) | this paper | BLW1834 | |
| Strain, strain background (*C. elegans*) | gfp::myrf-1 (1–482)::3xflag(syb1491) /mln1 II; glo-1(zu391) X; Pflp-13-mCherry::rab-3 (ybqIs1) | this paper | BLW1835 | |
| Strain, strain background (*C. elegans*) | gfp::myrf-1 (1–656)::3xflag(syb1468) /mln1 II; glo-1(zu391) X; Pflp-13-mCherry::rab-3 (ybqIs1) | this paper | BLW1836 | |
| Strain, strain background (*C. elegans*) | pan-1::gfp(ybq47) III | this paper | BLW1258 | |
| Strain, strain background (*C. elegans*) | pan-1::gfp(ybq47) III; glo-1(zu391) X | this paper | BLW1839 | |
| Strain, strain background (*C. elegans*) | pan-1::gfp(ybq47) III; glo-1(zu391) X; Pflp-13-mCherry::rab-3(ybqIs1) | this paper | BLW1878 | |
| Strain, strain background (*C. elegans*) | gfp::myrf-1::3xflag(ybq14) II; glo-1(zu391) X; Pflp-13-mCherry::rab-3(ybqIs1) | this paper | BLW1854 | |
| Strain, strain background (*C. elegans*) | gfp::myrf-1(ybq10) II | *Meng et al., 2017* | BLW831 | |
| Strain, strain background (*C. elegans*) | gfp::myrf-1::3xFlag(ybq14) II | *Meng et al., 2017* | BLW889 | |
| Strain, strain background (*C. elegans*) | gfp::myrf-2(ybq46) X | *Meng et al., 2017* | BLW1111 | |
| Strain, strain background (*C. elegans*) | gfp::myrf-1(ybq14)/mT1 II; pan-1(gk142)/mT1 III | this paper | BLW1465 | |
| Strain, strain background (*C. elegans*) | gfp::myrf-1::3xflag(ybq14)/ mT1 II; pan-1(gk142)/mT1 III | this paper | BLW1166 | |
| Strain, strain background (*C. elegans*) | gfp::myrf-1::3xflag(ybq14)/ mT1 II; pan-1(gk142)/mT1 III; Pscm-pan-1b::mCherry(ybqEx778) | this paper | BLW1731 | |
| Strain, strain background (*C. elegans*) | Pflp-13-GFP::rab-3 (ybqIs47) IV | this paper | BLW1419 | |
| Strain, strain background (*C. elegans*) | myrf-1(ju1121)/mln1 II; pan-1::GFP(ybq47) III | this paper | BLW1654 | |
| Strain, strain background (*C. elegans*) | gfp::myrf-1::3xflag(ybq14)/ mT1 II; pan-1(gk142)/mT1 III | this paper | BLW1883 | |

*Continued on next page*

*Appendix 1—key resources table continued*

| Reagent type (species) or resource | Designation | Source or reference | Identifiers | Additional information |
|---|---|---|---|---|
| Strain, strain background (*C. elegans*) | gfp::myrf-1 (1–700)(syb1313)/ mT1 II; pan-1(gk142)/mT1 III | this paper | BLW1805 | |
| Strain, strain background (*C. elegans*) | gfp::myrf-1 (1–790)(syb1333)/ mT1 II; pan-1(gk142)/mT1 III | this paper | BLW1806 | |
| Strain, strain background (*C. elegans*) | Pflp-13-GFP::rab-3(ybqIs47) III; Punc-25-pan-1(1-G513)::mCherry::KDEL(ybqEx815) | this paper | BLW1881 | |
| Strain, strain background (*C. elegans*) | Pflp-13-GFP::rab-3(ybqIs47) III; Punc-25-pan-1(1-G513)::mCherry::KDEL(ybqEx817) | this paper | BLW1890 | |
| Strain, strain background (*C. elegans*) | gfp::myrf-1::3xFlag(ybq14) II; Punc-25-pan-1(1-G513)::mCherry::KDEL(ybqEx807) | this paper | BLW1842 | |
| Strain, strain background (*C. elegans*) | gfp::myrf-1::3xFlag(ybq14) II; Pscm-Pan-1(1-G513)::mCherry::KDEL(ybqEx820) | this paper | BLW1895 | |
| Strain, strain background (*C. elegans*) | Punc-25-mCherry::Rab-3(juIs236)/mT1 II; pan-1(gk142)/mT1 III; Pflp-13-pan-1b:gfp (ybqEx821) | this paper | BLW1900 | |
| Strain, strain background (*C. elegans*) | pan-1(loxP)(syb1217) III; Pflp-13-gfp (juIs145)/Punc-25-mCherry::Rab-3(juIs236) II | this paper | BLW1825 | |
| Strain, strain background (*C. elegans*) | pan-1(loxP)(syb1217) III; Pflp-13-gfp(juIs145)/Punc-25-mCherry:: Rab-3(juIs236) II; Punc-25-Cre(tmIs1072) | this paper | BLW1784 | |
| Strain, strain background (*C. elegans*) | Pflp-13-GFP::rab-3(ybqIs47) III; Phlh-8-GFP(ayIs6) X | this paper | BLW1874 | |
| Strain, strain background (*C. elegans*) | Pflp-13-GFP::rab-3(ybqIs47) III; myrf-2(ybq42) Phlh-8-GFP(ayIs6) X | this paper | BLW1875 | |
| Strain, strain background (*C. elegans*) | gfp::myrf-1 (1–700)::3xflag(syb1313)/ mIn1 II; Pflp-13-GFP::rab-3(ybqIs47) III; Phlh-8-GFP(ayIs6) X | this paper | BLW1843 | |
| Strain, strain background (*C. elegans*) | gfp::myrf-1 (1–700)::3xflag(syb1313)/ mIn1 II; Pflp-13-GFP::rab-3(ybqIs47) III; myrf-2(ybq42) Phlh-8-GFP(ayIs6) X | this paper | BLW1848 | |
| Strain, strain background (*C. elegans*) | myrf-1(ybq6)/mIn1 II; myrf-2(ybq42) X; Pflp-13-GFP::rab-3(ybqIs47) III; Punc-47-gfp::myrf-1::flag, cb-unc-119(+)(ybqIs92(Si)) IV | this paper | BLW1781 | |
| Strain, strain background (*C. elegans*) | myrf-1(ybq6)/mIn1 II; myrf-2(ybq42) X; Pflp-13-GFP::rab-3(ybqIs47) III; Punc-47-gfp::myrf-1 (1–482)::flag, cb-unc-119(+)(ybqIs99(Si)) IV | this paper | BLW1782 | |
| Strain, strain background (*C. elegans*) | myrf-1(ybq6)/mIn1 II; myrf-2(ybq42) X; Pflp-13-GFP::rab-3(ybqIs47) III | this paper | BLW1008 | |
| Strain, strain background (*C. elegans*) | myrf-1(ybq6)/mIn1 II; Pflp-13-GFP::rab-3(ybqIs47) III | *Meng et al., 2017* | BLW827 | |

