## [Decision Letter]

**Acceptance summary:**

We appreciate how your paper describes an intriguing and novel mechanism for how a transcription factor is processed to affect gene regulation, and, as a consequence, synaptic connectivity.

**Decision letter after peer review:**

[Editors’ note: the authors submitted for reconsideration following the decision after peer review. What follows is the decision letter after the first round of review.]

Thank you for submitting your work entitled "The LRR-TM protein PAN-1 interacts with MYRF to promote its nuclear translocation in synaptic remodeling" for consideration by *eLife*. Your article has been reviewed by 3 peer reviewers, one of whom is a member of our Board of Reviewing Editors, and the evaluation has been overseen by a Senior Editor. The reviewers have opted to remain anonymous.

Our decision has been reached after consultation between the reviewers and editors. Based on these discussions and the individual reviews below, we regret to inform you that your work can not be considered for publication in *eLife* in its present form but are open to reconsideration if you are able to address the comments raised.

All reviewers very much appreciated the very interesting biological problem (DD neuronal rewiring) and were intrigued by the implication of a novel protein in the control of the very unusual MYRF transcription factor. The reviewers and editors also appreciated the many advances that you made, as well as the very solid nature of some the reagents that you used (particularly the CRISPR-generated alleles). However, all reviewers identified a substantial list of shortcomings. All these shortcomings are relatively straight-forward to address, but do amount to a substantial body of work. We will therefore need to reject the manuscript in its present form. However, should you be willing to engage with these specific requests, we may want to consider a substantially revised version of this manuscript sometime in the future.

Reviewer #1:

In this manuscript, the authors show molecular events underpinning rewiring of DD neurons in *C. elegans*. A previous study (Meng et al. 2017) showed that animals lacking the very unusual, but conserved transcription factor MYRF-1 show defects in DD rewiring at the late L1 stage. The present study finds that an LRR-containing transmembrane protein, PAN-1, interacts with the membrane bound MYRF-1. Both proteins localize to the cell membrane, and their interaction seems to be required for DD-rewiring. pan-1 mutants also show defects in DD-rewiring, and loss of PAN-1 affects MYRF-1 localization. The interaction between these two proteins occurs through their extracellular domains, and the extracellular portion of MYRF-1 is required for its activity.

I find this to be a very interesting paper that takes us a step further in understand how this very unusual activation mechanisms for a membrane bound (!) transcription factor works. However, some conclusions should be supported by more data and/or better data presentation. Moreover, some results are poorly described and there is general lack of detail throughout the manuscript. Specific points are listed below:

1. The manuscript shows that PAN-1 and MYRF-1 interact, and that mutants of each gene result in DD-rewiring defects (myrf-1 phenotype described previously in Meng et al. 2017). However, it does not adequately show that several perturbations leading to changes in PAN-1 and MYRF-1 localization result in DD-rewiring defects, the relevant paradigm of the study. For example, the authors make several truncations in PAN-1 and MYRF-1 to test their interaction using Co-IP (Figure 5), summarizing that their "results provide the first direct evidence that the non-cytoplasmic part of MYRF protein is indispensable for its function". However, they do not show what happens to DD-specific dorsal synapses in animals with truncations in MYRF-1 (Figures 5A and 6A).

2. The presentation of data for DD-rewiring defects in pan-1 mutants in Figures 1D, 1E and 5B is not convincing. Representative images in Figure 1C (middle) and 1F (bottom-right) clearly show that synapses are present, albeit much dimmer or with gaps, but this is not reflected in the quantification. Perhaps providing several representative images of pan-1 animals (and of pan-1a and pan-1c rescue in pan-1 mutants), like those shown in Figures 1F and S1A, would allow the reader to better interpret the DD-rewiring phenotype. Representative images should also be provided for PAN-1 variant animals quantified in Figure 5B.

3. Throughout the manuscript, the authors show changes in subcellular localization using seam cells as an example. These images are great and clearly convey the point, but serve as controls at best. Since the manuscript is framed around DD-rewiring, it is critical to show similar spatiotemporal changes in MYRF-1 localization in WT (Figure 3C) and pan-1 mutants (Figure 4E) in DD neurons. This region, with existing reporters, may not look as clear-cut, but an overall distribution of signal in cell-membrane / cytoplasm / nucleus may be evident using a confocal microscope and would allow the reader to better assess the results.

Reviewer #2:

The paper by Li et al. makes some intriguing findings about the mechanisms underlying DD synaptic remodeling in *C. elegans*. They find that PAN-1, a TM-LRR protein, can interact *in vitro* and *in vivo* with MYRF-1/2, a transmembrane-tethered DNA-binding protein previously shown to be required for DD remodeling. They present an interesting model in which PAN-1 localizes MYRF-1/2 to the plasma membrane (a novel finding) and that regulated nuclear entry of the MYRF-1/2 N-terminal fragment is important for its function. However, there are some gaps and over-interpretations that, for me, undermine confidence in the model.

1. For a few reasons, I have concerns about the specificity of the pan-1 phenotype. They don't show us whether overall DD anatomy is normal in pan-1 mutants. The DD/VD-specific rescue is nice, but isn't it possible that pan-1 expression in VDs (which are born during L1) has a role in DD rewiring? (This general concern applies to all of the experiments that use a promoter expressed in both neuron types.) Similarly, it's great to see the use of a pan-1 conditional allele, but its phenotype is really weak compared to the null, raising concern that some of the mutant defects could arise from pleiotropic functions (perhaps related to developmental delay/L2 arrest).

2. Along these lines, there's some confusion about what's meant by pan-1 and myrf-1 "activity." Because they're both expressed broadly, it's possible that they have different functions in different tissues and/or at different times. But the model doesn't consider this, and all phenotypes (whether in DDs, hypodermal cells, or larval arrest) are interpreted in terms of the same molecular model. Many of the key conclusions about what's going on in DD remodeling are drawn from looking in other cell types or at larval arrest (e.g. when the myrf-1 N-terminal fragment is overexpressed). This makes me less confident that we really understand what's going on in DDs themselves.

3. According to the simplest model, expression of the myrf-1 N-terminal fragment should cause premature DD remodeling. Unexpectedly, it causes early (L1) larval arrest, a gain-of-function phenotype, since the null arrests during L2. This is consistent with the idea that transient membrane localization of MYRF-1 is necessary, but it could also be that it's simply that temporally regulated production/activity of the N-terminal fragment is important. In either case, it seems important to know whether premature DD remodeling has occurred in the arrested L1s.

4. In the final set of experiments, the MYRF-1 extracellular region is overexpressed, which blocks synaptic remodeling. Authors conclude that this demonstrates that the MYRF-1-PAN-1 interaction is essential. However, many other interactions (or other process) could also be disrupted by overexpressing this fragment, so it's not clear that the defect is truly a specific consequence of the disrupted interaction.

Reviewer #3:

In a previous genetic screen, the authors identified the MYRF transcription factors as key-components required for synaptic rewiring of the DD neurons. By pulling down MYRF interacting partners, the authors here identified the membrane protein PAN-1 as a necessary factor required for MYRF activity. Previous models for MYRF activation suggested that MYRF undergo self-cleavage on ER membrane. The authors convincingly demonstrate that MYRF and PAN-1 physically and functionally interact, which is new and well documented, even if I have concerns with some data.

1. Because PAN-1 is found at the plasma membrane, the authors propose, first, that the localization at the PM is necessary for MYRF activation and, second, that "MYRF links cell surface activities to transcriptional cascades required for development". My main concern is that, at this stage, this remains an appealing hypothesis, which is not directly supported by experimental evidence. PAN-1 is also detected in intracellular compartments, at least in DD neurons (see Figure 2C), and might as well act by promoting MYRF oligomerization and cleavage early during the secretion pathway. Non-cleaved MYRF might then travel with PAN-1 to the membrane, but the functional relevance of this distribution is not demonstrated so far. One way to test this hypothesis would be to force the retention of PAN-1 in the ER and analyze MYRF processing and MYRF-dependent phenotypes.

2. In their previous article (Meng et al., Dev. Cell., 2017), the authors demonstrated that they could rescue DD rewiring by expressing the N' 1-482 domain of MYRF-1 in D neurons. In the present manuscript, they show that the same fragment no longer rescues myrf-1 (Figure 6E-H) and conclude that "MYRF membrane association is indispensable to its activity". This apparent contradiction needs to be solved.

3) At a technical level, imaging of PAN-1 and MYRF-1/2-GFP fusions are analyzed in a mixture of cell populations. Except for seam cells, the images are not very demonstrative and the quantification performed in the figures 3 and 4 appears quite subjective. To correlate DD rewiring with MYRF-1 relocalization, it should be necessary to focus on these neurons, use co-markers of ER, PM and nuclei, and perform quantitative analyses.

Altogether, it is not clear if PAN-1 is constitutively permissive for MYRF function in any tissue or can regulate MYRF activity, especially in the context of DD rewiring. Although a definite answer cannot be provided in this article, the first possibility cannot be dismissed and should be discussed.

---

## [Author Response]

[Editors’ note: the authors resubmitted a revised version of the paper for consideration. What follows is the authors’ response to the first round of review.]

All reviewers very much appreciated the very interesting biological problem (DD neuronal rewiring) and were intrigued by the implication of a novel protein in the control of the very unusual MYRF transcription factor. The reviewers and editors also appreciated the many advances that you made, as well as the very solid nature of some the reagents that you used (particularly the CRISPR-generated alleles). However, all reviewers identified a substantial list of shortcomings. All these shortcomings are relatively straight-forward to address, but do amount to a substantial body of work. We will therefore need to reject the manuscript in its present form. However, should you be willing to engage with these specific requests, we may want to consider a substantially revised version of this manuscript sometime in the future.

Common key points:

1. While all new myrf-1 truncation alleles exhibit larval arrest, there is a lack of characterization of DD rewiring phenotype in those mutants.

We have now added the analysis of DD synaptic rewiring for all new myrf-1 truncation alleles, and corresponding myrf-1(truncation); myrf-2(null) double mutants. All new myrf-1 alleles, except for myrf-1(syb1313), behave similarly to previously characterized myrf-1 loss of function mutants, i.e., they exhibit blocked synaptic rewiring in myrf-2 null background. The exception is myrf-1(syb1313)^1-700 GFP^, in which the whole extracellular region of MYRF-1 is deleted. myrf1(syb1313) exhibits precocious yet discordant DD rewiring. We have also observed that MYRF-1^1-700 GFP^(syb1313) can be partially processed in ER to produce N-MYRF while the majority of MYRF-1^1-700 GFP^(syb1313) protein appears to be in ER. MYRF-1^1-700 GFP^(syb1313) signals remained strong in pan-1(0) mutants, while the signals of MYRF-1^1-790 GFP^(syb1333), in which part of extracellular region was removed, were undetectable in pan-1(0), resembling wild type MYRF in pan-1(0). Thus, from an unexpected angle, these new results support that the extracellular region of MYRF is indispensable for MYRF’s processing, and the cleavage of wild type MYRF is under developmental regulation. See Figure 6; 8, 9; Figure S5-16.

2. There is a lack of localization analysis for truncated MYRF-1^GFP^ in DD neurons.

MYRF-1-GFP signals are very weak, e.g., they cannot be detected by conventional confocal. Fortunately, we were able to image them under Zeiss Airyscan. For certain imaging experiments, myrf-1 alleles were crossed into glo-1 mutant background to reduce gut auto fluorescence. We have added new sets of images to display signal patterns of MYRF-1 variants. We have also added images of colabeling MYRF-1^GFP^ variants with DD-specifically expressed mCherry::Rab-3. Rab-3 signals are localized to presynaptic vesicles, but also often in cytoplasm, presumably labeling a subset of ER, Golgi, or endosomes. With contrastive localization of MYRF-1^GFP^ variants and mCherry::Rab-3, one can appreciate the differential localization of MYRF-1^GFP^ variants on cell membrane, cytoplasm, and nucleus. Because DDs are small in size (3_µ_m in diameter), further colocalization analysis using specific organelle markers are often unproductive due to the resolution limitation. See Figure 3D, 4F, 5G, 8F, 12G.

3. The choice of flp-13 and unc-25 promoters

unc-25 encodes Glutamic Acid Decarboxylase, and the particular unc-25 promoter that we use labels 4 RMEs (head neurons that form neuromuscular junctions with head muscles to control foraging) and 6 DDs in L1 larvae. The unc-25 promoter also labels 13 VDs, which are born at late L1. flp-13 encodes a FMRF-Like Peptide, and is expressed in 6 DDs not in VDs among ventral cord motor neurons; however, flp-13 is expressed in many other head neurons. We even observed that the flp-13 promoter-driven marker was expressed in intestine, and such expression is particularly strong during early-mid L1, which can obscure the DD signals. flp-13pro is often weak in DD5 and very weak in DD6. flp-13_pro_ driven synaptic marker also label an unidentified head neuron that extend its axon to the anterior half of ventral cord, which only becomes clear until L3-L4. Together, unc-25 and flp-13 have its pro and con as synaptic marker for DDs.

When it comes to experiments of cell-autonomy test, unc-25 is overall a more restricted promoter than flp-13, since flp-13 is expressed much more broadly than unc-25. Take an example of pan-1 rescue experiments, we found that many flp-13_pro_-pan-1 worms were thin and pale, and difficult to form lines, while unc25_pro_-pan-1 worms did not show any signs of sickness. We reasoned that the overexpression of pan-1 caused defects because flp-13_pro_ was expressed in vital tissues.

Even though the birth of VDs coincides with DD rewiring in timing, VDs are not critical for DD rewiring, because lineage mutants that do not generate VDs exhibits normal DD rewiring (Hallam and Jin, 1998).

We have added analysis for DD rewiring phenotype of pan-1(0) using flp-13_pro_ marker. We have added pan-1 rescue experiments use flp-13_pro_ driven pan-1. The results were consistent with the conclusion based on unc-25_pro_. See Figure S1A.

We used flp-13_pro_ synpatic marker for analyses of DD rewiring in all myrf-1 truncation alleles. See Figure 6; Figure S5-16.

4. Phenotypic discrepancy between N-MYRF-1^1-482^ exchromosomal array transgene and myrf-1^1-482^(syb1491) allele

Previously, MYRF-1^1-482^ was overexpressed in multi-copy exchromosomal arrays, and it caused early DD rewiring (Meng, 2017). In contrast, the present myrf-1^1-482^(syb1491) allele exhibit deficient DD rewiring.

One notable difference between the two experiments was that the expression level driven by exchromosomal array was much higher than endogenous level, considering that the transgene GFP::MYRF-1^1-482^ can be readily detected in nucleus (e.g., by 10x objective lens), while MYRF-1^1-482^(syb1491) was very weak (e.g., by 60x oil objective lens and prolonged camera exposure).

We have performed new experiment in which we generated single copy insertion transgene to express full length MYRF-1 and MYRF-1^1-482^. We found that full length MYRF-1 rescued the rewiring defects in myrf-1(0); myrf-2(0) double mutants, while MYRF-1^1-482^ did not. This results support the hypothesis that the low expression of N-MYRF is insufficient for DD rewiring.

Studies of mammalian MYRF show that monomer N-MYRF does not function as efficiently as trimer N-MYRF. When full length MYRF is trimerized and cleaved, the N-MYRF is released as form of trimer. The C-tail of N-MYRF is required for N-MYRF maintain trimer form (Kim D. 2017 Nucleic Acids Res).

It is conceivable that the membrane localization is necessary for MYRF (of endogenous level) to oligomerize, because the 2-dimension greatly increases the probability of interaction between protomers.

MYRF-1^1-700^(syb1313) (with no extracellular domain) can indeed generate N-MYRF that are sufficient for early rewiring in a subset of DDs, even though the cleavage process is inefficient. Comparing to MYRF-1^1-700^(syb1313), MYRF-1^1656^(syb1468) is only short of TMD. MYRF-1^1-656^(syb1468) produced plenty of NMYRF-1 (which is likely as monomer) but the rewiring is deficient. This support that membrane association greatly facilitates the cleavage of MYRF.

We have included the data as Figure S21, and discussed the results. See page 28, line 569.

5. DD rewiring defects show incomplete penetrance in pan-1^loxp^; unc-25_pro_-Cre.

We added an illustration of DD anatomy in Figure 1C. There are processes from six DDs tiled along ventral and dorsal cords. If only one or two DDs are defective in rewiring, one or two segments of synapses would appear absent in the dorsal cord.

We consider that the rewiring defect in pan-1^loxp^; unc-25_pro_-Cre is comparable to pan-1(0) mutants. We observed a sustained rewiring block by L4 (in Figure S1C) in these animals, which indicates the defect is not secondary to developmental arrest.

The incomplete penetrance in DD rewiring was likely because the effects of Cre was not penetrant among individual DDs. Cre is driven by unc-25 promoter, and unc-25 is reported to be expressed during late embryogenesis. Signals of DD rewiring can be detected as early as 16 post-hatch hours. This concludes that the time window between Cre transcription and DD rewiring is about 18 hours. The duration may not be long enough to allow Cre to excise the target with high penetrance. See page 11, line 209.

Reviewer #1:In this manuscript, the authors show molecular events underpinning rewiring of DD neurons in *C. elegans*. A previous study (Meng et al. 2017) showed that animals lacking the very unusual, but conserved transcription factor MYRF-1 show defects in DD rewiring at the late L1 stage. The present study finds that an LRR-containing transmembrane protein, PAN-1, interacts with the membrane bound MYRF-1. Both proteins localize to the cell membrane, and their interaction seems to be required for DD-rewiring. pan-1 mutants also show defects in DD-rewiring, and loss of PAN-1 affects MYRF-1 localization. The interaction between these two proteins occurs through their extracellular domains, and the extracellular portion of MYRF-1 is required for its activity.I find this to be a very interesting paper that takes us a step further in understand how this very unusual activation mechanisms for a membrane bound (!) transcription factor works. However, some conclusions should be supported by more data and/or better data presentation. Moreover, some results are poorly described and there is general lack of detail throughout the manuscript. Specific points are listed below:1. The manuscript shows that PAN-1 and MYRF-1 interact, and that mutants of each gene result in DD-rewiring defects (myrf-1 phenotype described previously in Meng et al. 2017). However, it does not adequately show that several perturbations leading to changes in PAN-1 and MYRF-1 localization result in DD-rewiring defects, the relevant paradigm of the study. For example, the authors make several truncations in PAN-1 and MYRF-1 to test their interaction using Co-IP (Figure 5), summarizing that their "results provide the first direct evidence that the non-cytoplasmic part of MYRF protein is indispensable for its function". However, they do not show what happens to DD-specific dorsal synapses in animals with truncations in MYRF-1 (Figures 5A and 6A).

See our response to Common Key Point #1.

2. The presentation of data for DD-rewiring defects in pan-1 mutants in Figures 1D, 1E and 5B is not convincing. Representative images in Figure 1C (middle) and 1F (bottom-right) clearly show that synapses are present, albeit much dimmer or with gaps, but this is not reflected in the quantification. Perhaps providing several representative images of pan-1 animals (and of pan-1a and pan-1c rescue in pan-1 mutants), like those shown in Figures 1F and S1A, would allow the reader to better interpret the DD-rewiring phenotype. Representative images should also be provided for PAN-1 variant animals quantified in Figure 5B.

We have added new figures, in which multiple ventral, dorsal cords images were vertically tiled. To represent the true synaptic pattern of the animals, the display contrast of ventral and dorsal cord images was adjusted to the same parameters for comparison. The counting of synapse was performed using Find Peaks, a plug-in program in ImageJ. The parameters were set in a way to faithfully detect all clearly visible synapses in wild type.

We agree that there were dim, small, puncta present in the dorsal cords of the mutants. It is of our interest to further characterize those structures. We consider that the synaptic rewiring occurs in multiple steps, and plan to analyze how myrf and pan-1 may be involved in specific steps.

3. Throughout the manuscript, the authors show changes in subcellular localization using seam cells as an example. These images are great and clearly convey the point, but serve as controls at best. Since the manuscript is framed around DD-rewiring, it is critical to show similar spatiotemporal changes in MYRF-1 localization in WT (Figure 3C) and pan-1 mutants (Figure 4E) in DD neurons. This region, with existing reporters, may not look as clear-cut, but an overall distribution of signal in cell-membrane / cytoplasm / nucleus may be evident using a confocal microscope and would allow the reader to better assess the results.

See our response to Common Key Point #2.

Reviewer #2:The paper by Li et al. makes some intriguing findings about the mechanisms underlying DD synaptic remodeling in *C. elegans*. They find that PAN-1, a TM-LRR protein, can interact *in vitro* and *in vivo* with MYRF-1/2, a transmembrane-tethered DNA-binding protein previously shown to be required for DD remodeling. They present an interesting model in which PAN-1 localizes MYRF-1/2 to the plasma membrane (a novel finding) and that regulated nuclear entry of the MYRF-1/2 N-terminal fragment is important for its function. However, there are some gaps and over-interpretations that, for me, undermine confidence in the model.1. For a few reasons, I have concerns about the specificity of the pan-1 phenotype. They don't show us whether overall DD anatomy is normal in pan-1 mutants. The DD/VD-specific rescue is nice, but isn't it possible that pan-1 expression in VDs (which are born during L1) has a role in DD rewiring? (This general concern applies to all of the experiments that use a promoter expressed in both neuron types.) Similarly, it's great to see the use of a pan-1 conditional allele, but its phenotype is really weak compared to the null, raising concern that some of the mutant defects could arise from pleiotropic functions (perhaps related to developmental delay/L2 arrest).

We have added new data that axon morphology of DDs was normal in pan-1 mutants and in pan-1^loxp^; unc-25_pro_-Cre animals. See Figure 1F.

See our response to Common Key Point #3 for choice of promoters and VD’s role in DD rewiring.

See our response to Common Key Point #5 about the penetrance of Cre.

Our results indicate that pan-1 is required cell-autonomously in DD rewiring. Expression of pan-1 by unc-25_pro_ and flp-13_pro_ (two promoters intersect at DDs) in pan-1(0) rescued DD rewiring; in other words, presence of pan-1 in DDs sufficiently promotes DD rewiring in an arrested animal. This implies that pan-1 specifically regulates DD rewiring.

We agree that pan-1 regulates development of other cell types. Certain mechanisms by which pan-1 regulates DD rewiring may apply to development of non-DD cells.

In our research, we did not observe direct link between animal arrest and defective DD rewiring. On the contrary, we have seen a great number of L2arrested mutants (many do not have VD), but the DD rewiring was normal. A good example is myrf-1(0) single mutant; it arrests at the end of L1 with largely normal DD rewiring. In contrast, myrf-1(0); myrf-2(0) double mutants can develop further (to the end of L2), but the rewiring is blocked.

Based on our screening, the arrest (late L1 or L2) mutants are many, while rewiring-blocked mutants are extremely rare.

2. Along these lines, there's some confusion about what's meant by pan-1 and myrf-1 "activity." Because they're both expressed broadly, it's possible that they have different functions in different tissues and/or at different times. But the model doesn't consider this, and all phenotypes (whether in DDs, hypodermal cells, or larval arrest) are interpreted in terms of the same molecular model. Many of the key conclusions about what's going on in DD remodeling are drawn from looking in other cell types or at larval arrest (e.g. when the myrf-1 N-terminal fragment is overexpressed). This makes me less confident that we really understand what's going on in DDs themselves.

See our response to Common Key Point #1.

3. According to the simplest model, expression of the myrf-1 N-terminal fragment should cause premature DD remodeling. Unexpectedly, it causes early (L1) larval arrest, a gain-of-function phenotype, since the null arrests during L2. This is consistent with the idea that transient membrane localization of MYRF-1 is necessary, but it could also be that it's simply that temporally regulated production/activity of the N-terminal fragment is important. In either case, it seems important to know whether premature DD remodeling has occurred in the arrested L1s.

See our response to Common Key Point #1 and #4.

4. In the final set of experiments, the MYRF-1 extracellular region is overexpressed, which blocks synaptic remodeling. Authors conclude that this demonstrates that the MYRF-1-PAN-1 interaction is essential. However, many other interactions (or other process) could also be disrupted by overexpressing this fragment, so it's not clear that the defect is truly a specific consequence of the disrupted interaction.

We agree that there are more than one possibilities to explain the dominant negative effects of MYRF-1^681-931^ overexpression. We took Reviewer #3’s suggestion and performed a new experiment in which an ER-retained PAN-1 was over expressed under unc-25_pro_. The transgene caused severely deficient DD rewiring. The ER-retained pan-1 also trapped MYRF^GFP^ in ER as MYRF^GFP^ signals were greatly increased in DDs comparing to those in neighboring cells. These results support that PAN-1 interacting MYRF begins in ER, but their interaction merely in ER is insufficient for MYRF processing.

We are further analyzing interaction between PAN-1 and MYRF by protein domain mapping and structural analysis. With additional insights, more precise perturbation of MYRF-PAN-1 interaction could be designed.

Reviewer #3:In a previous genetic screen, the authors identified the MYRF transcription factors as key-components required for synaptic rewiring of the DD neurons. By pulling down MYRF interacting partners, the authors here identified the membrane protein PAN-1 as a necessary factor required for MYRF activity. Previous models for MYRF activation suggested that MYRF undergo self-cleavage on ER membrane. The authors convincingly demonstrate that MYRF and PAN-1 physically and functionally interact, which is new and well documented, even if I have concerns with some data.1. Because PAN-1 is found at the plasma membrane, the authors propose, first, that the localization at the PM is necessary for MYRF activation and, second, that "MYRF links cell surface activities to transcriptional cascades required for development". My main concern is that, at this stage, this remains an appealing hypothesis, which is not directly supported by experimental evidence. PAN-1 is also detected in intracellular compartments, at least in DD neurons (see Figure 2C), and might as well act by promoting MYRF oligomerization and cleavage early during the secretion pathway. Non-cleaved MYRF might then travel with PAN-1 to the membrane, but the functional relevance of this distribution is not demonstrated so far. One way to test this hypothesis would be to force the retention of PAN-1 in the ER and analyze MYRF processing and MYRF-dependent phenotypes.

We overexpressed ER-retained pan-1 under unc-25 promoter in myrf1^GFP^(ybq14). The transgene caused severely deficient DD rewiring. The ERretained pan-1 also trapped MYRF^GFP^ in ER as MYRF^GFP^ signals were greatly increased in DDs comparing to those in neighboring cells. These results support that PAN-1 interacting MYRF begins in ER, but the ER-contained MYRF-PAN-1 interaction is insufficient for MYRF processing. See Figure 11B-D.

We agree that further mechanisms remain to be determined.

We have changed “MYRF links cell surface activities…” to “MYRF may link cell surface activities..”. See page 1, summary, line 33.

2. In their previous article (Meng et al., Dev. Cell., 2017), the authors demonstrated that they could rescue DD rewiring by expressing the N' 1-482 domain of MYRF-1 in D neurons. In the present manuscript, they show that the same fragment no longer rescues myrf-1 (Figure 6E-H) and conclude that "MYRF membrane association is indispensable to its activity". This apparent contradiction needs to be solved.

See our response to Common Key Point #4.

3. At a technical level, imaging of PAN-1 and MYRF-1/2-GFP fusions are analyzed in a mixture of cell populations. Except for seam cells, the images are not very demonstrative and the quantification performed in the figures 3 and 4 appears quite subjective. To correlate DD rewiring with MYRF-1 relocalization, it should be necessary to focus on these neurons, use co-markers of ER, PM and nuclei, and perform quantitative analyses.

See our response to Common Key Point #1 and #2.

Altogether, it is not clear if PAN-1 is constitutively permissive for MYRF function in any tissue or can regulate MYRF activity, especially in the context of DD rewiring. Although a definite answer cannot be provided in this article, the first possibility cannot be dismissed and should be discussed.

We have added the sentences in discussion: “Our results demonstrate that PAN-1 is required for proper function of MYRF. One possibility is PAN-1 is constitutively permissive for MYRF function, e.g., PAN-1 stabilizes the extracellular domain of MYRF; however, aberrant processing and function of MYRF^1-700 GFP^ indicate that the extracellular domain of MYRF is indispensable, further suggesting a regulatory role for PAN-1.” See page 26, line 531.